# Extreme purifying selection against point mutations in the human genome

Noah Dukler[1,3], Mehreen R. Mughal[1,3], Ritika Ramani[1], Yi-Fei Huang [2] & Adam Siepel [1✉]

Large-scale genome sequencing has enabled the measurement of strong purifying selection in protein-coding genes. Here we describe a new method, called ExtRaINSIGHT, for measuring such selection in noncoding as well as coding regions of the human genome. ExtRa-INSIGHT estimates the prevalence of "ultraselection" by the fractional depletion of rare single-nucleotide variants, after controlling for variation in mutation rates. Applying ExtRa-INSIGHT to 71,702 whole genome sequences from gnomAD v3, we find abundant ultra-selection in evolutionarily ancient miRNAs and neuronal protein-coding genes, as well as at splice sites. By contrast, we find much less ultraselection in other noncoding RNAs and transcription factor binding sites, and only modest levels in ultraconserved elements. We estimate that ~0.4–0.7% of the human genome is ultraselected, implying ~ 0.26–0.51 strongly deleterious mutations per generation. Overall, our study sheds new light on the genome-wide distribution of fitness effects by combining deep sequencing data and classical theory from population genetics.

[1] Simons Center for Quantitative Biology, Cold Spring Harbor Laboratory, Cold Spring Harbor, NY, USA. [2] Department of Biology and Huck Institute of the Life Sciences, The Pennsylvania State University, University Park, PA, USA. [3] These authors contributed equally: Noah Dukler, Mehreen R. Mughal.
✉email: asiepel@cshl.edu

L ike a gambler, an evolving species has to pay for the chance to win. As in most games of chance, the majority of "draws" (mutations) result in a loss (decrease in fitness), with an occasional pay-off (adaptive mutation). Thus, in Haldane's words, loss of fitness owing to deleterious mutation is the "price paid by a species for its capacity for further evolution"[1].

Understanding the impact of new mutations on fitness has been a major focus of evolutionary genetics for at least a century[1–3], with implications for a wide variety of fundamental problems, ranging from revealing the genetic architecture of complex traits and the effects of mutational load to understanding the emergence of recombination and sex[4,5]. Nevertheless, it is notoriously difficult to characterize the full distribution of fitness effects (DFE) of new mutations. Naturally occurring mutations are rare, often difficult to detect, and have fitness effects that are generally hard to measure. Innovative experimental techniques have been developed to measure the DFE in model organisms, but these methods have important limitations[4] and, in any case, they cannot be applied to humans, nor to any other organism that cannot be experimentally manipulated and monitored in relatively large numbers.

For these reasons, many recent efforts to characterize the DFE have focused on the study of naturally occurring mutations using statistical modeling, population genetic theory, and DNA sequencing[6–9]. Patterns of genetic variation are strongly influenced by demographic history, however, so careful demographic modeling is required to isolate the effects of selection. In addition, most available population panels—consisting of hundreds to a few thousand individuals—are informative about only a relatively narrow slice of the DFE. For example, in humans strong purifying selection (such that $s > {\sim}1\%$) will tend to hold variants below a detectable frequency in these panels, whereas weak purifying selection (such that $s < {\sim}10^{-4}$) will be indistinguishable from random genetic drift[10,11]. Thus, only in approximately the range $10^{-4} < s < 10^{-2}$ can purifying selection be accurately measured.

Recently, exome or whole-genome sequence data has become available for tens of thousands of individuals[12,13], allowing quite rare variants (with relative frequencies $< 10^{-3}$) to be identified with reasonable confidence. These data have enabled the application of statistical methods that can measure high levels of purifying selection against predicted loss-of-function (pLoF) mutations for protein-coding genes by comparing the frequencies of pLoF variants to their mutation-rate-based expectation[11–16]. For example, the widely used "probability of being loss-of-function intolerant" (pLI) measure, and its successor, the "loss-of-function observed/expected upper bound fraction" (LOEUF) measure, have been shown to reliably distinguish among null (unconstrained), autosomal recessive, and haploinsufficient genes[12,13].

While such measures are correlated with dominance effects, the frequency of rare pLoF variants is strictly informative only about the strength of selection against hetereozygous mutations, $s_{\mathrm{het}}$[17]. Indeed, if purifying selection is strong, near-complete recessivity can be excluded, and mutation-selection balance holds, then the equilibrium frequency for a rare variant should occur at $q \approx \frac{\mu}{s_{\mathrm{het}}}$, where $\mu$ is the deleterious mutation rate[1,17]. Cassa et al.[11] (see also[18]) have argued that this relationship holds quite well for pLoF variants in the ExAC exome data[12] from large values of $s_{\mathrm{het}}$ down to $s_{\mathrm{het}} \approx 0.01$ (but see ref.[19]). Importantly, estimation of $s_{\mathrm{het}}$ based on mutation-selection balance is independent of demography because, in this regime, mutant alleles persist in the population for at most a few generations and genetic drift makes a negligible contribution to their allele frequencies.

In this article, we extend and generalize these ideas for application to the entire genome, including noncoding regions, in a new method called Extremely Rare INSIGHT (ExtRaINSIGHT). Similar to our previous Inference of Natural Selection from Interspersed Genomically coHerent elemenTs (INSIGHT) method[20,21], ExtRaINSIGHT can be used to measure the influence of natural selection on any designated set of genomic sequences, by contrasting patterns of variation in a designated set of "target" sequences with those in matched sequences that are putatively neutrally evolving. However, ExtRaINSIGHT focuses on rare variants only, in order to obtain a measure that reflects particularly large selective effects—that is, purifying selection sufficiently strong that new point mutations do not appear even as rare variants in a panel of tens of thousands of individuals. As shorthand, we refer to such selection as "ultraselection." ExtRaINSIGHT does not directly estimate $s_{\mathrm{het}}$ but rather a parameter, denoted $\lambda_s$, that represents the fractional depletion of rare variants owing to purifying selection. However, we show that, if mutation-selection balance can be assumed and $\lambda_s$ is sufficiently large, approximate estimates of $s_{\mathrm{het}}$ can be obtained based on a simple relationship with $\lambda_s$. We apply ExtRaINSIGHT to more than 70,000 whole genome sequences from the Genome Aggregation Database (gnomAD) project (https://gnomad.broadinstitute.org/)[13] and perform a comprehensive analysis of ultraselection in the human genome, considering both coding and noncoding elements. Our findings reveal both similarities and striking differences in measures of ultraselection and weaker purifying selection, shed light on the rate of strongly deleterious mutations in humans, and highlight challenges in accurately modeling mutation rates in upstream regions of genes.

## Results

**Overview of ExtRaINSIGHT**. ExtRaINSIGHT measures the fractional reduction in the incidence of rare variants in a target set of sites relative to nearby sites that are putatively free from (direct) natural selection. In this way, it is analogous to classical strategies for measuring selection in protein-coding genes[22–24], as well as to newer methods that compare target sets of noncoding elements with suitable background sequences[21,25–27]. The focus on rare variants (here, variants with minor allele frequencies of $< 0.1\%$), however, enables the method to focus in particular on point mutations of large selective effect.

The main challenge in this approach stems from the high sensitivity of relative rates of rare variants to variation in mutation rate. To address this problem, we follow refs.[12,15] in building a mutational model that accounts for both sequence context and regional variation in mutation rate. In our case, we condition the rate of each type of nucleotide substitution on the identity of the three flanking nucleotides on each side. In addition, following our earlier work[20,21], we use a local control for overall mutation rate based on nearby sites identified as likely to be neutrally evolving. We also consider G+C content, sequencing coverage, and CpG islands as covariates (see Methods). With this strategy, we are able to predict with high accuracy the probability that a rare variant will occur at each site (Supplementary Fig. 1). Notably, this mutation model is also predictive of *de novo* variants from ref.[28] (Supplementary Fig. 3), which should be even less influenced by selection than the rare variants in gnomAD.

In the absence of natural selection, we assume a Bernoulli sampling model for the presence (probability $P_i$) or absence (probably $1 - P_i$) of a rare variant at each site $i$, where $P_i$ reflects the local sequence context and overall rate of mutation. We ignore sites at which common variants occur (similar to refs.[12,15]). We then assume that natural selection has the effect of imposing a fractional reduction on the rate at which rare variants occur. To a

first approximation, we maximize the following likelihood function,

$$
\begin{aligned}
\mathcal{L}(\lambda_s; \mathbb{Y}, \mathbb{P}) &= P(\mathbb{Y}; \lambda_s, \mathbb{P}) \\
&= \prod_i \left[(1 - \lambda_s)P_i\right]^{Y_i} \left[1 - (1 - \lambda_s)P_i\right]^{1 - Y_i}
\end{aligned}
\tag{1}
$$

where $Y_i$ is an indicator variable for the presence of a rare variant at position $i$ in the sample, $\lambda_s$ is a scale factor capturing a depletion of rare genetic variation, $\mathbb{Y} = \{Y_i\}$, $\mathbb{P} = \{P_i\}$, and the product excludes sites having common variants. By maximizing this function we can obtain a maximum-likelihood estimate (MLE) of $\lambda_s$ conditional on pre-estimated values $P_i$. (In practice, we use a slightly more complicated likelihood function that distinguishes among the possible alternative alleles at each site; see "Methods" for complete details.) Assuming the $P_i$ values are pre-estimated, an approximate, unbiased maximum-likelihood estimator (MLE) for $\lambda_s$ and an estimator for its variance can be obtained in closed form (see "Methods"). Importantly, this variance has almost no sensitivity to variance in the pre-estimated $P_i$ values in the regime of interest (see Supplementary Fig. 4), making the model highly robust to uncertainty in mutation rate estimates provided they are unbiased.

When $\lambda_s$ falls between 0 and 1 it can be interpreted as a measure of the prevalence of ultraselection. In this case, $\lambda_s$ can be thought of as the fraction of sites intolerant to heterozygous mutations, although in practice, some sites may be more, and some sites less, intolerant. Notice, however, that $\lambda_s$ can also take values < 0 if rare variants occur at a higher-than-expected rate in the target set of sites. As we discuss below, we do observe a systematic tendency for $\lambda_s$ to take negative values in particular classes of sites, likely reflecting the difficulty of precisely specifying the mutational model at these sites. Across most of the genome, however, estimates of $\lambda_s$ fall between 0 and 1 and show general qualitative agreement with other measures of purifying selection.

Notably, in the case of strong selection against heterozygotes and mutation-selection balance (as detailed by refs. [11,17]), a relatively simple relationship can be established between $\lambda_s$ and the site-specific selection coefficient against heterozygous mutations, $s_{het}$ (see Eq. (12) in "Methods" and Supplementary Fig. 5). To test this relationship, following ref. [18], we simulated data sets under a realistic human demographic model with various values of $s_{het}$ and estimated $\lambda_s$ from each one. We found that this approach led to highly accurate estimates of the true value down to about $s_{het} = 0.03$, and somewhat elevated but acceptable estimates down to about $s_{het} = 0.02$ (Supplementary Fig. 6), which corresponds to $\lambda_s \approx 0.45$ with our data set. As it turns out, most of our estimates from real data do not exceed this threshold but when they do, we use this approach to estimate $s_{het}$. Importantly, it is only these approximate estimates of $s_{het}$, not $\lambda_s$ itself, that depend on the assumption of mutation-selection balance.

**Ultraselection in and around protein-coding genes**. We applied ExtRaINSIGHT to 19,955 protein-coding genes from GENCODE v. 38 [29] as well as to a variety of proximal coding-associated sequences, including 5′ and 3′ untranslated regions (UTRs), promoters, and splice sites (Fig. 1). For comparison, we applied INSIGHT to the same sets of elements. As expected, we obtained considerably higher estimates of $\lambda_s$ at 0-fold degenerate (0d) sites in coding sequences, at which each possible mutation results in an amino-acid change ($\lambda_s = 0.22$), than at 4-fold degenerate (4d) sites, at which every mutation is synonymous ($\lambda_s = -0.008$). The corresponding INSIGHT-based estimates of $\rho$ were 0.80 and 0.39, respectively. Together, we can interpret these estimates as

indicating that 22% of 0d sites are ultraselected, meaning that any mutation at these sites would be strongly deleterious, and another $80 - 22 = 58\%$ are under weaker purifying selection—although the ExtRaINSIGHT and INSIGHT estimates are not precisely comparable in all respects (see "Discussion"). By contrast, at 4d sites, ultraselection is estimated to be completely absent, but 39% of 4d sites experience weak purifying selection (see ref. [9] for an estimate of 26% for synonymous sites). Overall, about 15% of coding sites (CDS) experience ultraselection ($\lambda_s = 0.15$) and another 47% experience weaker selection ($\rho = 0.62$).

Among coding-related sites, the strongest selection, by far, occurred in splice sites (see also ref. [30]), where almost half of sites were subject to ultraselection ($\lambda_s = 0.45$; corresponding to $s_{het} \approx 0.02$), with another 43% subject to weaker selection ($\rho = 0.88$). By contrast, 3′ UTRs showed little evidence of ultraselection ($\lambda_s = 0.028$) despite considerable evidence of weaker selection ($\rho = 0.24$). Interestingly, we observed a persistent tendency for negative estimates of $\lambda_s$ at regions near the 5′ ends of genes, at both 5′ UTRs and promoter regions, despite non-neglible estimates of $\rho$ (0.22 and 0.13, respectively). As we discuss in a later section, these estimates appear to be a consequence of unusual mutational patterns in these regions that are difficult to accommodate using even our regional and neighbor-dependent mutation model.

To see whether ExtRaINSIGHT was capable of distinguishing among protein-coding sequences experiencing different levels of selection against heterozygous loss-of-function (LoF) variants, we compared it with the recently introduced "loss-of-function observed/expected upper bound fraction" (LOEUF) measure[13]. LOEUF is similarly based on rare variants but differs from ExtRaINSIGHT in that it is computed separately for each gene by pooling together all mutations predicted to result in loss-of-function of that gene (including nonsense mutations, mutations that disrupt splice sites, and frameshift mutations). In contrast to $\lambda_s$ and $\rho$, lower LOEUF scores are associated with stronger depletions of LoF variants and increased constraint, and higher LOEUF scores are associated with weaker depletions and reduced constraint. To compare the two measures, we partitioned 80,950 different isoforms of 19,677 genes into deciles by LOEUF score and ran ExtRaINSIGHT separately on the pooled coding sites corresponding to each decile. Again, we computed $\rho$ values using INSIGHT together with the $\lambda_s$ values. We found that both $\rho$ and $\lambda_s$ decreased monotonically with LOEUF decile, with $\lambda_s$ ranging from 0.28 for the genes having the lowest LOEUF scores to 0.008 for the genes having the highest LOEUF scores, and $\rho$ similarly ranging from 0.77 to 0.43 (Fig. 1). These results suggest that in the 10% of genes under the weakest selection against heterozygous LoF mutations, only 0.8% of sites are subject to ultraselection, but over 40% still experience weaker purifying selection; whereas in the 10% of genes under the strongest selection against LoF mutations, almost 30% of sites are under ultraselection and another ~ 40% are under weaker purifying selection.

Finally, we considered an alternative grouping of genes by biological pathway, using the top-level annotation from the Reactome pathway database[31] (Fig. 2). Again, we ran both ExtRaINSIGHT and INSIGHT on each group of genes and observed similar trends in the two measures, with $\lambda_s$ ranging from 10% to 27%, and $\rho$ ranging from 61% to 75%. We found genes annotated as belonging to the "Neuronal System" to be experiencing the most ultraselection ($\lambda_s = 0.27$), consistent with other recent findings[9]. Genes annotated as being involved in "Reproduction" showed the least ultraselection ($\lambda_s = 0.10$). Notably, the estimates of $\lambda_s$ exhibited considerably greater variation, as a fraction of the mean, than did estimates of $\rho$. The ratio $\lambda_s/\rho$—which can be interpreted as the fraction of

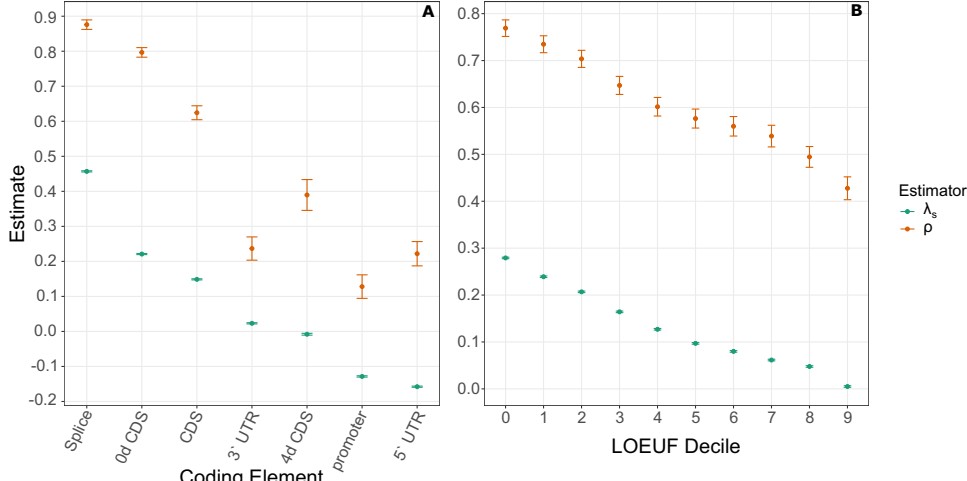

**Fig. 1 Measures of purifying selection at coding and coding-proximal genomic elements.** **A** Estimates for various annotation types are shown for both ExtRaINSIGHT ($\lambda_s$; teal) and INSIGHT ($\rho$; orange). **B** Similar estimates are shown for protein-coding genes by deciles of the loss-of-function observed/expected upper bound fraction (LOEUF) measure[13]. Results are shown for 80,950 isoforms of 19,677 genes. Notice that lower LOEUF scores are associated with stronger depletions of LoF variants, so $\lambda_s$ and $\rho$ tend to decrease as LOEUF increases. Error bars are centered at the MLE and indicate one standard error in each direction (see "Methods").

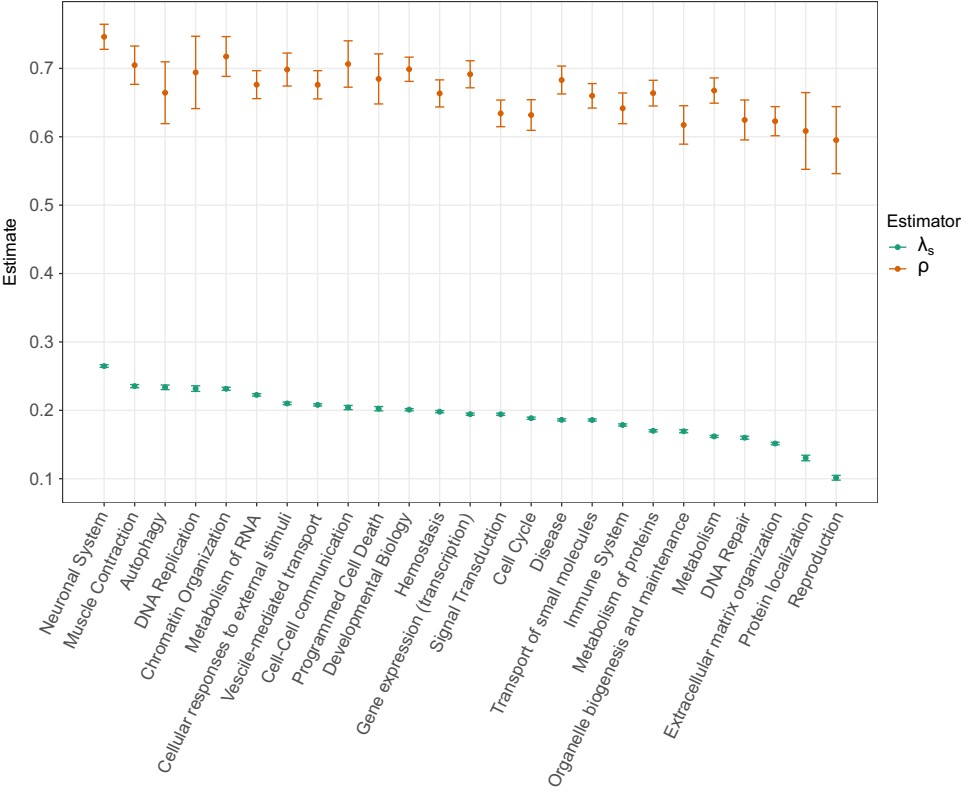

**Fig. 2 Measures of purifying selection in protein-coding genes by biological pathway.** Genes were assigned coarse-grained functional categories using the top-level annotation from the Reactome pathway database[31]. An estimates for each category is shown for both ExtRaINSIGHT ($\lambda_s$; teal) and INSIGHT ($\rho$; orange). Error bars are centered at the MLE and indicate one standard error in each direction (see "Methods"). Total number of genes: $n = 19,256$ (ranging from 125 to 2707 per category).

selected sites experiencing ultraselection—was also highest for "Neuronal System" genes (at 0.36) and lowest for "Reproduction" genes (at 0.18). An analysis of genes exhibiting tissue-specific expression produced similar results, with several brain tissues exhibiting the most ultraselection and vagina exhibiting the least (Supplementary Fig. 7).

**Ultraselection in noncoding elements.** We carried out a similar analysis on noncoding sequences, including a variety of noncoding RNAs, transcription factor binding sites (TFBS) supported by chromatin-immunoprecipitation-and-sequencing (ChIP-seq) data (from ref. [21]), and unannotated intronic and intergenic regions. Among these sequences, we observed the strongest

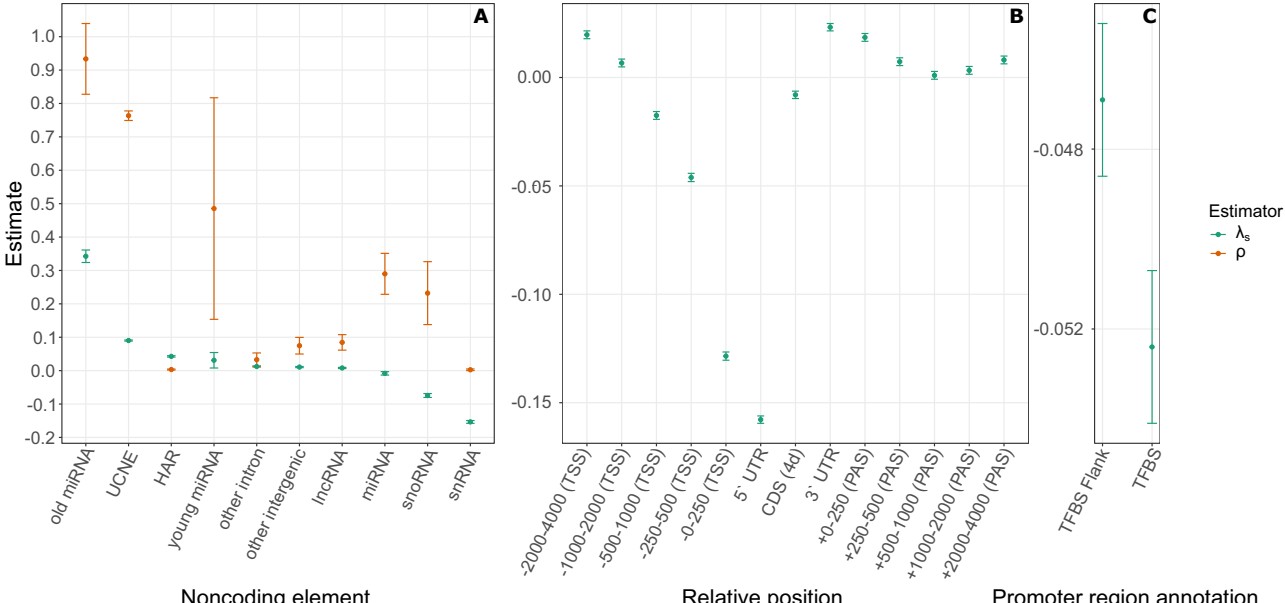

**Fig. 3 Measures of purifying selection at annotated noncoding elements and in genomic intervals near protein-coding genes. A** Estimates for both ExtRaINSIGHT ($\lambda_s$; teal) and INSIGHT ($\rho$; orange) at noncoding elements (x-axis). **B** Estimates of $\lambda_s$ in windows upstream of the transcription start site (TSS) and downstream of the polyadenylation site (PAS) (x-axis). The 5′ and 3′ UTRs are also shown, as are fourfold degenerate (4d) coding sites (CDS). **C** Estimates of $\lambda_s$ for the extended promoter region (2kb upstream of the TSS) within transcription factor binding sites (TFBS) annotated in the Ensembl Regulatory Build[44] and in the immediate flanking sequences (10bp on each side). The difference in (**C**) is highly statistically significant by a two-sided likelihood ratio test based on the ExtRaINSIGHT likelihood model ($p = 2.8 \times 10^{-13}$). Error bars are centered at the MLE and indicate one standard error in each direction (see "Methods"). Numbers of elements: **A** old miRNA: $n = 7537$; UCNE: $n = 1,415,142$; HAR: $n = 674,492$; young miRNA: $n = 6285$; other intron: $n = 971,109,276$; other intergenic: $n = 1,255,478,347$; lncRNA: $n = 453,200,392$; miRNA: $n = 140,681$; snoRNA: $n = 49,837$; snRNA: $n = 155,304$; **B** $n = 58,496$. **C** $n = 1,120,839$.

signature of ultraselection in microRNAs (miRNAs), particularly in evolutionarily "old" miRNAs broadly shared across mammals (designated as "conserved" by TargetScan; see "Methods"), where we estimated $\lambda_s = 0.34$ (Fig. 3). We found that the seed regions of these miRNAs had even slightly higher values of $\lambda_s = 0.39$. Interestingly, however, the prevalance of ultraselection was greatly reduced at evolutionarily "new" miRNAs that are not shared across mammals ("nonconserved" in TargetScan), where we estimated only $\lambda_s = 0.031$.

Other types of noncoding RNAs also showed little indication of ultraselection: our estimates for long noncoding RNAs (lncRNAs), small nuclear RNAs (snRNAs), and small nucleolar RNAs (snoRNAs) were all close to zero or negative. In an attempt to identify regions within these RNAs that might be subject to stronger selection, we intersected them with conserved elements identified by phastCons[25]. However, we found that even these putatively conserved portions of noncoding RNAs exhibited at most $\lambda_s \approx 0.05$ (in lncRNAs).

When we analyzed a pooled set of all ~ 2M TFBSs from ref. [21], we obtained a negative estimate of $\lambda_s = -0.08$, despite that the same elements yielded a nonnegligible estimate of $\rho = 0.23$. We therefore examined only the binding sites of the 10 TFs whose binding sites showed the largest $\rho$ estimates ($\rho = 0.61$ overall; see "Methods"), but even for this putatively conserved set, we obtained an estimate of only $\lambda_s = 0.03$. Thus, of the noncoding RNA and TFBSs we considered, only "old" miRNAs appear to experience high levels of ultraselection.

We also evaluated ultraconserved noncoding elements (UCNEs)[32] and noncoding human accelerated regions (HARs)[33–35]—two types of elements that have been widely studied for their unusual patterns of cross-species conservation, and have been shown to function in various ways, including as enhancers[36,37] and noncoding-RNA transcription units[33].

Interestingly, despite their extreme levels of cross-species conservation, UCNEs show only modest levels of ultraselection, with $\lambda_s = 0.09$. This observation suggests that what is unusual about these elements is not the strength of selection acting on them (which is considerably weaker than that at protein-coding sequences or "old" miRNAs), but instead the uniformity of selection acting at each nucleotide (see "Discussion"). Notably, HARs display only slightly lower levels of ultraselection than UCNEs ($\lambda_s = 0.04$) and levels comparable to those of conserved sequences in introns. Thus, despite their rapid evolutionary change during the past 5–7 million years, HARs now appear to contain many nucleotides that are under strong purifying selection in human populations.

**A genome-wide accounting of sites subject to ultraselection.** To account genome-wide for the incidence of strongly deleterious mutations, we ran ExtRaINSIGHT on a collection of mutually exclusive and exhaustive annotations. For this analysis, we considered CDSs, UTRs, splice sites, lncRNAs, introns, and intergenic regions, but excluded smaller classes of noncoding RNAs, which make negligible genome-wide contributions (Table 1). As above, we intersected the lncRNA, intron, and intergenic classes with phastCons elements, and separately considered the conserved and nonconserved partitions of each class. For each category, we multiplied our estimate of $\lambda_s$ by the number of sites in the category to estimate category-specific expected numbers of sites subject to ultraselection. To account for potential misspecification of the mutational model, we conservatively subtracted from the category-specific estimates of $\lambda_s$ the estimate for nonconserved intronic regions (0.009). Thus, by construction, the expected number of ultraselected sites in these and similar regions (including nonconserved intergenic and lncRNA sites) was zero.

**Table 1 Ultraselection across the human genome (based on ExtRaINSIGHT).**

| Feature | $\lambda_s$ | ± (stderr)[a] | no. sites (M) | prop. sites | exp. no. (M)[b] | exp. prop.[c] | fold enrich. | exp. s-del.[d] | $s_{het}$ |
|---|---|---|---|---|---|---|---|---|---|
| CDS | 0.148 | 0.0004 | 33.8 | 1.18% | 4.7 | 43.5% | 36.9 | 0.11 | – |
| 5′ UTR | −0.161 | 0.0006 | 8.2 | 0.29% | 0.0 | 0.0% | 0.0 | 0.00 | – |
| 3′ UTR | 0.028 | 0.0002 | 36.1 | 1.26% | 0.7 | 6.2% | 5.0 | 0.02 | – |
| splice | 0.464 | 0.0012 | 0.8 | 0.03% | 0.4 | 3.3% | 121.3 | 0.01 | 2.0% |
| nonconserved lncRNA[e] | 0.009 | 0.0001 | 453.6 | 15.78% | 0.0 | 0.0% | 0.0 | 0.00 | – |
| conserved lncRNA[f] | 0.055 | 0.0003 | 23.3 | 0.81% | 1.1 | 9.8% | 12.1 | 0.03 | – |
| nonconserved intron[e] | 0.009 | 0.0000 | 972.6 | 33.83% | 0.0 | 0.0% | 0.0 | 0.00 | – |
| conserved intron[f] | 0.058 | 0.0002 | 44.3 | 1.54% | 2.2 | 20.1% | 13.1 | 0.05 | – |
| nonconserved intergenic[e] | 0.003 | 0.0000 | 1255.5 | 43.67% | 0.0 | 0.0% | 0.0 | 0.00 | – |
| conserved intergenic[f] | 0.048 | 0.0002 | 46.9 | 1.63% | 1.8 | 17.0% | 10.5 | 0.04 | – |
| Total | | | 2875.1 | 100.00% | 10.8 | 100.0% | | 0.26 | |

[a]The similar values of the standard errors (equal after rounding) reflect the maximum of 1M sites used for estimation.
[b]Expected number of ultraselected sites after adjusting for background. In this case, the estimate for nonconserved introns (0.009) was subtracted from each estimate of $\lambda_s$ (see Supplementary Table 1 for a less conservative correction).
[c]Expected proportion of ultraselected sites after adjusting for background.
[d]Expected number of new strongly deleterious mutations per diploid individual, assuming a mutation rate of $1.2 \times 10^{-8}$ per generation per site.
[e]Sites not classified as conserved by phastCons.
[f]Sites classified as conserved by phastCons.

Overall, we estimated that 0.374% ± 0.002% of the human genome is ultraselected, with 44% of ultraselected sites falling in CDSs, 13% in conserved introns, 11% in conserved intergenic regions, 12% in conserved lncRNAs, 5% in 3′ UTRs and 3% in splice sites. Notably, ultraselected sites are overrepresented 37-fold in CDSs, but CDSs still account for less than half of ultraselected sites. Splice sites are overrepresented 121-fold but make a minor overall contribution owing to their small number.

Our assumption is that any point mutation at these ultraselected sites will be strongly deleterious, and simulations indicate that the detected sites are indeed subject to extreme purifying selection (see Discussion). Thus, if we multiply the expected numbers of sites by twice (allowing for heterozygous mutations) the estimated per-generation, per-nucleotide mutation rate (here assumed to be $1.2 \times 10^{-8}$ ref. [38]), we obtain expected numbers of de novo strongly deleterious mutations per potential zygote ("potential" because some mutations will act prior to fertilization). By this method, we estimate 0.258 ± 0.001 strongly deleterious mutations per potential zygote. By construction, these strongly deleterious mutations occur in the same category-specific proportions as the ultraselected sites (44% from CDS, 23% from introns, etc.). Thus, we expect about 0.11 strongly deleterious coding mutations per potential zygote and about another 0.15 such mutations at various noncoding sites.

If we carry out a less conservative version of these calculations, by subtracting the $\lambda_s$ estimate for nonconserved intergenic regions (0.003) rather than the one for intronic regions, we estimate 0.732% ± 0.004% of the genome to be ultraselected, with 23% falling in CDSs (Supplementary Table 1). The expected number of strongly deleterious mutations per potential zygote increases to 0.505 ± 0.003, of which 0.12 fall in CDSs. Taking these calculations together, we estimate a range of 0.26–0.51 strongly deleterious mutations per potential zygote, implying a high genetic burden but one that appears to be roughly compatible with other lines of evidence (see "Discussion").

We performed a parallel analysis using INSIGHT, to estimate the numbers and distribution of more weakly deleterious mutations (Table 2). In this case, we estimate that 3.2% of sites are under selection and the expected number of de novo deleterious mutations per fertilization is 2.21. The fraction of deleterious mutations from CDS is 22%, with most of the remainder coming from introns and intergenic regions. lncRNAs

and 3′ UTRs also make significant contributions. Taking the ExtRaINSIGHT and INSIGHT estimates together, we estimate that each potential fertilization event is associated with 0.26–0.51 new strongly deleterious mutations and an additional 1.70–1.95 new mutations that are more weakly deleterious. One way to interpret these numbers is that, conditional on a threshold level of fitness (i.e., the existence of no strongly deleterious mutations), each person contains an expected ~2 new mutations that are sufficiently deleterious that they would tend to be eliminated from the population on the time-scale of human-chimpanzee divergence (as measured by INSIGHT), at least if humans continued to experience historical levels of purifying selection. That person's genetic load would derive from both these new mutations and similar weakly deleterious mutations passed down from his or her ancestors.

**Local misspecification of the mutation model.** As noted above, we observed a consistent tendency to estimate negative values of $\lambda_s$ at the 5′ ends of genes, including in 5′ UTRs and core promoters (Fig. 1), as well as at TFBSs and some noncoding RNAs from across the genome (Fig. 3). In an attempt to bound the genomic regions near protein-coding genes that give rise to these negative estimates, we applied ExtRaINSIGHT in a series of windows near the 5′ and 3′ ends of genes, pooling data from all ~ 20,000 genes (Fig. 3b). We found that the effect was most pronounced in the 5′ UTR, where we estimated $\lambda_s = -0.16$ (see Fig. 1) and in the 250bp immediately upstream of the TSS ($\lambda_s = -0.13$). As we looked farther upstream, it diminished fairly rapidly, with $\lambda_s = -0.05$ in the $(-500, -250)$ window and $\lambda_s = -0.02$ in the $(-1000, -500)$ window. By the $(-2000, -1000)$ window, the estimates had returned to slightly positive values. We did not observe negative estimates near the 3′ ends of genes, and the estimate for 4d sites within the CDS was only slightly negative. Therefore, the tendency to estimate $\lambda_s < 0$ near genes appears to be limited to the 5′ UTR and the ~1 kb region upstream of the TSS.

We hypothesized that, despite being well-calibrated across the majority of the genome (Supplementary Fig. 1), our mutation model is misspecified in promoter regions, perhaps owing to correlations of mutation rates with features such as chromatin accessibility or hypomethylation. We therefore adapted our model to consider the predicted state from an application of the 25-state ChromHMM model[39,40] to Roadmap Epigenomics

**Table 2 Weaker selection across the human genome (based on INSIGHT).**

| Feature | $\rho$ | ± (stderr) | no. sites (M) | prop. sites | exp. no. (M)[a] | exp. prop.[b] | fold enrich. | exp. del.[c] |
|---|---|---|---|---|---|---|---|---|
| CDS | 0.624 | 0.020 | 33.8 | 1.18% | 19.7 | 21.5% | 18.2 | 0.47 |
| 5′ UTR | 0.222 | 0.035 | 8.2 | 0.29% | 1.5 | 1.6% | 5.6 | 0.04 |
| 3′ UTR | 0.237 | 0.033 | 36.1 | 1.26% | 7.0 | 7.7% | 6.1 | 0.17 |
| splice | 0.883 | 0.013 | 0.8 | 0.03% | 0.7 | 0.7% | 26.3 | 0.02 |
| nonconserved lncRNA[d] | 0.025 | 0.020 | 453.6 | 15.78% | 0.0 | 0.0% | 0.0 | 0.00 |
| conserved lncRNA[e] | 0.412 | 0.019 | 23.3 | 0.81% | 8.6 | 9.4% | 11.6 | 0.21 |
| nonconserved intron[d] | 0.042 | 0.022 | 972.6 | 33.83% | 0.0 | 0.0% | 0.0 | 0.00 |
| conserved intron[e] | 0.426 | 0.019 | 44.3 | 1.54% | 17.0 | 18.5% | 12.0 | 0.41 |
| nonconserved intergenic[d] | 0.059 | 0.036 | 1255.5 | 43.67% | 21.7 | 23.6% | 0.5 | 0.52 |
| conserved intergenic[e] | 0.376 | 0.020 | 46.9 | 1.63% | 15.7 | 17.0% | 10.4 | 0.38 |
| Total | | | 2875.1 | 100.00% | 91.9 | 100.0% | | 2.21 |

[a]Expected number of deleterious sites after adjusting for background. In this case, the estimate for nonconserved introns (0.022) was subtracted from each estimate of $\rho$.
[b]Expected proportion of deleterious sites after adjusting for background.
[c]Expected number of new deleterious mutations per diploid individual, assuming a mutation rate of $1.2 \times 10^{-8}$ per generation per site.
[d]Sites not classified as conserved by phastCons.
[e]Sites classified as conserved by phastCons.

data[41] as a categorical covariate and refitted it to the data, trying ChromHMM predictions for several cell types. However, we found that this approach did not eliminate the tendency for negative estimates of $\lambda_s$, perhaps because the available epigenomic data has too coarse a resolution or is not well matched by cell type.

Having observed negative estimates of $\lambda_s$ also at TFBSs outside of promoter regions, however, we wondered if the effect could be driven, at least in part, by TF binding itself, which has been shown to be mutagenic in melanoma[42,43]. In an attempt to isolate the effects of TF binding, we applied ExtRaINSIGHT separately to predicted TFBS in extended promoter regions, using predictions from the Ensembl Regulatory Build[44], and to the immediate flanking 10bp on either side of these predictions, excluding flanking sequences that themselves included TFBSs. Interestingly, we found that estimates of $\lambda_s$ were significantly more negative in the TFBSs than in the immediate flanking sites (Fig. 3c); $p = 2.8 \times 10^{-13}$, likelihood ratio test), suggesting a possible influence from the mutagenic effects of TF binding (see "Discussion"). In the end, we were not able to eliminate this apparent problem with our mutation model, but its effects appear to be generally quite local to TSSs and TFBSs and therefore are likely to have a limited impact on our genome-wide analyses.

## Discussion
In this article, we have introduced a new method, called ExtRa-INSIGHT, for measuring the prevalence of strong purifying selection, or "ultraselection," on any collection of sites in the human genome, including noncoding as well as coding sites. ExtRaINSIGHT enables maximum-likelihood estimation of a parameter, denoted $\lambda_s$, that represents the fractional depletion in rare variants in a target set of sites relative to matched "neutral" sites, after accounting for neighbor-dependence and local variation in mutation rate. We have surveyed the prevalence of ultraselection in both coding and non-coding regions of the human genome and found it to be particularly strong in splice sites, 0-fold degenerate (0d) coding sites, and evolutionarily ancient miRNAs. On the other hand, ultraselection is mostly absent in other noncoding RNAs, untranslated regions of protein-coding genes, and transcription factor binding sites, as well as in fourfold degenerate (4d) coding sites. We have also shown that neural-related genes and genes expressed in the brain are enriched for large estimates of $\lambda_s$ in their coding sequences, whereas reproduction-related genes are enriched for small estimates of $\lambda_s$.

Perhaps the most challenging aspect of our analysis is fully accounting for variation in mutation rate, so that our estimates of $\lambda_s$ truly reflect the action of purifying selection alone. We made use of a model that accounts for several known correlates of true or apparent mutation rate, including neighboring nucleotides, genomic position, G+C content, and sequencing coverage. We also excluded CpGs entirely, owing to their highly atypical mutational patterns. Overall, we found that our mutation model provides a good fit to the observed numbers of rare variants in putatively neutral regions (Supplementary Fig. 1; see also Supplementary Fig. 3), but we did find that some classes of sites display clear excesses of rare variants (Supplementary Fig. 2). The clearest example of this phenomenon was the promoter regions of genes, consistent with our tendency to observe negative estimates of $\lambda_s$ in these regions (as discussed further below), although we also observed slight excesses in repetitive regions. When we exclude repeats and promoter regions, the observed numbers of rare variants match our model reasonably well, in terms of both the mean and the variance (Supplementary Fig. 1). Importantly, as far as we can tell, the misspecification of our model always seems to result in an under-prediction, rather than an over-prediction, of the number of rare variants under neutrality, which will tend to make our estimates of $\lambda_s$ conservative. In addition, we find that our estimator for $\lambda_s$ is highly insensitive to variance in the sitewise mutation rates, as long as they are unbiased (Supplementary Fig. 4). Therefore, some overdispersion of mutation rates relative to our model should have a negligible effect on our analysis, as long as the sites in a target class do not tend to be skewed in the same direction. For these reasons, we have not attempted to extend our model to explicitly account for overdispersion, as in studies of somatic mutations in cancer[45,46], although this could be an area worth exploring in future work.

While our study focuses primarily on $\lambda_s$, a measure of depletion of rare variants, we also show that when $\lambda_s$ is sufficiently large (approximately $> 0.45$ for our data) and mutation-selection balance is assumed, $1 - \lambda_s$ is expected to have an inverse relationship with the selection coefficient against heterozygous mutations, which allows $s_{het}$ to be approximately estimated for a target collection of sites. Simulations indicate that this approximation is reasonably good when selection is strong and uniform, although it is biased upward near the boundary of $\lambda_s \approx 0.45$ (Supplementary Fig. 5). In addition, when selection is variable across sites this estimator will describe the harmonic mean, rather than the arithmetic mean, of the true values (see "Methods", Supplementary Fig. 6). Consequently, it will have a predictable downward bias, meaning that it can be interpreted as a lower-bound on the true arithmetic mean. For these reasons, we focus our analysis primarily on $\lambda_s$ and use corresponding estimates of $s_{het}$ only for

context and interpretation when $\lambda_s$ is sufficiently large. It is worth emphasizing that our estimates of $\lambda_s$ do not depend on the assumption of mutation-selection bias. These estimates do, however, have a quantitative dependence on the size of the data set and subjective choices regarding the allele-frequency threshold for rare variants and the criteria for putatively neutral sequences, among other features.

Interestingly, we found only a modest prevalence of ultraselection in ultraconserved noncoding elements (UCNEs), despite their near-complete sequence conservation over hundreds of millions of years of evolution[32]. It has been suggested that this extreme conservation is indicative of strong purifying selection (e.g., ref. [32]), although most such observations have not been accompanied by direct estimation of selection coefficients. One exception is an early study by Katzman et al.[47], where ultraconserved elements in humans were estimated to be experiencing substantially stronger selection (by about 3-fold) than non-synonymous sites in protein-coding sequences, although the absolute strength of selection was estimated to be modest (mean of $2N_e s \approx -5$) and the analysis was based on only 72 individuals. The assumption of strong levels of selection has been difficult to reconcile with observations that organisms often appear to function normally after deletion of UCNEs, as when complete deletion of several UCNEs in mice failed to produce detectable phenotypes[48] (see also ref. [49]). More recently, Snetkova et al. found that UCNEs were remarkably resilient to mutation, with a majority continuing to function as enhancers in transgenic mouse reporter assays even after being subjected to substantial levels of mutagenesis[50]. Our observations suggest that these apparently contradictory observations—high sequence conservation and resilience to mutation—can be reconciled if UCNEs are predominantly under relatively weak selection, that is, selection strong enough to prohibit fixation of new mutations on the time scales of interspecies divergence but weak enough that rare variants are not substantially depleted. Our simulations suggest that values of $s_{het}$ between about 0.003 and 0.005 result in such behavior (Supplementary Fig. 8). Indeed, we find considerably lower levels of ultraselection in UCNEs ($\lambda_s = 0.09$) than in 0d sites in coding regions ($\lambda_s = 0.22$) or in ancient miRNAs ($\lambda_s = 0.34$). At the same time, these other classes of sites tend not to show perfect conservation in cross-species comparisons, primarily because they tend to be interspersed with less conserved sites (e.g., 4d sites or non-pairing sites in miRNAs). Thus, what seems to be most unusual about UCNEs is not the extreme level of purifying selection they experience but rather the uniformity of purifying selection across hundreds of bases and across many different species. In most cases it is still unknown what causes this uniformity, although it has been speculated that it may result from overlapping functional roles, such as overlapping binding sites, structural RNAs, and coding regions[32].

It is instructive to compare our estimates of $\lambda_s$ in and around protein-coding genes with previous estimates of the DFE for these regions. Our estimate of $\lambda_s = 0.45$ for splice sites corresponds to $s_{het} \approx 0.02$, which is reasonably concordant with Cassa et al.'s[11] mean estimate of $s_{het} = 0.059$ for predicted loss-of-function (pLoF) variants in protein-coding genes, assuming that many but not all splice-site-disrupting mutations result in loss of function, and allowing for our possible under-estimation of $s_{het}$ in the presence of variability across sites. However, our estimate of $\lambda_s = 0.22$ for missense mutations at 0d sites appears to be somewhat larger than expected in comparison to studies based on the site-frequency-spectrum[5–8]. For example, the best-fitting such model in a representative recent study by Kim et al.[8], based on a fairly large sample size (432 Europeans from the 1000 Genomes Project), implied a mean selection coefficient against amino-acid replacements of $s_{het} = 0.007$. If we apply ExtRaINSIGHT to data

simulated under Kim et al.'s DFE, we obtain an estimate of only $\lambda_s = 0.08$, or about one third of our estimate of $\lambda_s = 0.22$ for real 0d sites (Supplementary Table 2, Supplementary Fig. 9). Thus, the patterns of rare variants present in the deeply sequenced gnomAD data set do not seem to be consistent with the DFEs inferred from smaller data sets. Our methods do not allow for estimates of $s_{het}$ in these regions (because $\lambda_s$ is too low), but this discrepancy in $\lambda_s$ estimates from the real and simulated data suggests that the SFS-based methods have under-estimated the weight of the tail of the DFE, which is well known to be difficult to measure based on the SFS particularly with samples of modest size (e.g., ref. [7]).

A possible concern with our approach is that, in estimating $\lambda_s$ from the rare variants missing from the target sites, ExtRaINSIGHT inevitably will pick up not only on strongly deleterious mutations but also, to a degree, on selection on a large class of more weakly deleterious mutations. Even if these more weakly deleterious mutations are inefficiently eliminated over the short time scale relevant for rare variants, their cumulative effect could still be substantial relative to that from strongly deleterious mutations if they are much larger in number—which is plausible if the weight in the tail of the true DFE is not too large. Such a scenario could potentially lead to overestimation of $\lambda_s$ and, consequently, of $s_{het}$ and of the numbers of strongly deleterious mutations per potential fertilization.

We attempted to examine this question by simulating data under four different DFEs, representing scenarios from quite weak selection to quite strong selection, applying ExtRaINSIGHT to the simulated data, and then decomposing the DFE into a component associated with the rare variants removed by selection and a component associated with the remaining rare variants (which we can trace in simulation; see Supplementary Fig. 9 and Supplementary Table 2). The first simulated DFE was based on the model inferred by Kim et al.[8] for coding regions, and the other three were adapted from it to generate values of $\lambda_s$ similar to what we observed in coding regions, evolutionary ancient miRNAs, and TFBSs (Supplementary Table 2). We found, overall, that the missing variants detected by ExtRaINSIGHT are heavily enriched for strong purifying selection. In the case of quite strong selection, they predominantly have $s_{het} > 0.01$, with mean values of $s_{het}$ ranging from 0.016–0.027. Even in the case of Kim et al.'s inferred DFE (which, as discussed above, may under-estimate the tail), the mean $s_{het} = 0.016$ for the missing rare variants, although in this case substantially more of them have $s_{het} < 0.01$. Overall, we find that, with mean $s_{het} \approx 0.02$, these rare variants are indeed under quite strong purifying selection, although our power to separate strong and weak purifying selection does depend on the original DFE.

Throughout this article, we have compared $\lambda_s$ estimates from ExtRaINSIGHT with $\rho$ estimates from INSIGHT, in order to evaluate the relative fractions of sites subject to ultraselection and weaker forms of purifying selection. It is worth noting, however, that the two methods are not based on precisely the same assumptions and therefore are not exactly comparable. Unlike ExtRaINSIGHT, INSIGHT measures natural selection on the time scale of the human-chimpanzee divergence (5–7 MY), assuming that functional roles are relatively constant during that time period. It also incorporates positive selection as well as purifying selection into its model, although positive selection appears to make at most a minor contribution to $\rho$ in this setting (see "Methods"). Finally, INSIGHT makes use of a much simpler Jukes-Cantor mutation model, with no accounting for neighbor-dependence in mutation rate (although it does account for regional variation across the genome). As a result, differences between $\lambda_s$ and $\rho$ could result in part from matters such as gain and loss of functional elements on human/chimp time scales,

misspecification of the Jukes-Cantor mutation model, or contributions from positive selection. Nevertheless, we expect these differences to have relatively minor effects, and the estimates from INSIGHT and ExtRaINSIGHT appear to be fairly consistent overall, with $\rho$ and $\lambda_s$ well correlated but $\rho > \lambda_s$ in all cases. Therefore, we believe it is reasonable to approximately characterize the DFE by treating $\lambda_s$ as a measure of ultraselection and the difference $\lambda_s - \rho$ as a measure of selection that is weaker but sufficiently strong to result in removal of deleterious variants on the time scale of human/chimpanzee divergence.

What are the implications of our estimates of ~ 0.26–0.51 for the number of strongly deleterious mutations and of ~ 2 more weakly deleterious mutations per diploid genome per generation? These estimate imply a fairly high genetic burden but one that appears to be in the plausible range. For comparison, Eyre-Walker and Keightley[51] estimated 1.6 (±0.8) deleterious mutations per generation for coding regions only based on a comparison with the chimpanzee genome; Morten et al.[52] estimated 3–5 lethal equivalents for the entire genome based on consanguineous marriages; and Muller[53] estimated 0.2–1.0 de novo deleterious mutations per diploid genome per generation, which would correspond to a range of 0.9–4.5 based on a modern estimate of the number of human genes[30]. Notably, our estimate is depressed by our conservative correction for model misspecification, which results in a prediction that only 3.2% of the genome is under selection, compared with our previous INSIGHT-based estimate of 4.2–7.5%[54] and an alternative estimate of 8.2%[55]. A less conservative correction could increase our estimate for the total number of deleterious mutations by as much as a factor of 2.5, bringing it more in line with some of the larger previous estimates. Another rough point of comparison is the rate of spontaneous abortion, which has been estimated to be as high as 50% for mothers of prime reproductive age[56,57]. This quantity, of course, is not directly comparable to the estimates of deleterious mutations per generation for a variety of reasons but the observation is consistent with a fairly high mutational load. It is worth recalling that, according to classical arguments[1,24,53], estimates of greater than one lethal equivalent per fertilization are inconsistent with population survival under a model where each mutation makes an independent contribution to reduction in fitness.

Despite several attempts, we were not able to eliminate the apparent misspecification of our mutation model in promoter regions as well as at other TFBSs and at some noncoding RNAs. This misspecification is unlikely to be explained by unusual base or word composition in these regions, nor by regional variation in overall mutation rate, because these features are explicitly addressed by our model. We also could not eliminate it by explicitly conditioning on chromatin state, using the ChromHMM model[39,40], although it is possible that our approach was limited by the resolution and cell-type-specificity of the available epigenomic data. Interestingly, the best predictor we could identify for elevated mutation rates was TF binding itself. There is accumulating evidence from melanoma that TF binding may be mutagenic, likely because it interferes with DNA repair[42,43], so it seems possible that TF binding is, at least in part, a driver of elevated germ-line mutation rates in these regions. It is worth noting that if TF binding indeed itself significantly alters mutation rates, this phenomenon would considerably complicate efforts to measure natural selection on TFBS, which is generally accomplished by contrasting rates of polymorphism and/or divergence within binding sites relative to nearby flanking sites, under the assumption that mutation rates are approximately equal in these regions (e.g., refs. [21,27,58]). However, the strength of this mutagenic effect in the germline remains unknown, and unless it is particularly pronounced, it likely has a minor effect on

analyses at longer evolutionary time scales, where natural selection probably dominates in determining patterns of polymorphism and divergence. In any case, more work will be needed to develop a full understanding of these potential mutational biases and account for them in analyses of selection on binding sites.

## Methods

**Data for neutral model.** The data for our neutral model consisted of rare variants (MAF < 0.001) from gnomAD (v3) within the genomic regions identified by Arbiza et al.[21] as putatively free from selection, unduplicated, non-repetitive, and reliably mappable. These regions were mapped to the hg38 human assembly using liftOver[59]. We further removed all CpG sites, which we expected to be difficult to model owing to methylation-induced hypermutation, and all sites having an an average sequencing coverage across individuals of <20 reads.

**Mutation model.** To fit the mutation model to these putatively neutral sites, we first calculated the relative frequencies of each type of mutation $a \to b$ and of the absence of a mutation ($a \to a$), conditional on the identities of $a$, $b$, and the three flanking nucleotides on each side. This required collecting $4^8 = 65536$ distinct counts (minus the excluded CpGs) and normalizing them to sum to one separately for each $a$ and flanking nucleotides. We then obtained adjusted rates by combining the (logits of) these raw relative rates with a collection of covariates likely to be correlated with real or apparent rates of mutation in a linear-logistic model. In particular, we used four covariates: the raw relative frequency, the logarithm of the reported average sequencing coverage from gnomAD, the fractional G+C content in a 200bp window, and an indicator for whether or not each site fell in a CpG island (based on the UCSC Genome Browser track of the same name[59]). We fitted this model to the observed rates of mutation at variable and nonvariable sites, sampling 1% of putatively neutral sites for efficiency. Finally, we further adjusted the estimated rates for regional variation in mutation rate by sliding a 150kb window along the genome in 50kb increments, and fitting a linear-logistic model to the neutral sites in each window, with the logit of the previously estimated rate as a covariate with coefficient one and a free intercept term, which could be interpreted as a local scaling factor. Together, these steps allowed us to estimate an absolute rate for the emergence of each allele at each site in the genome. When we compare the predicted rates with actual rates within the neutral regions, we can see that the model is quite well calibrated (Supplementary Fig. 1).

To validate our mutation model, we quantified the occurrence of de novo mutations and compared them to the predicted probability of mutation. Each de novo variant characterized in ref. [28] includes the site at which the mutation occurred and the specific allele change. We first mapped these variants from hg19 to hg38 using liftOver[59], resulting in 174,122 mapped mutations. Using this information we mapped each de novo variant to the probability of observing that specific mutation according to our model. We counted the number of de novo variants that occurred conditional on ranges of predicted mutation rate. Comparing these counts to the predicted mutations rates, we observed a clear correlation (Supplementary Fig. 3).

**Approximate model for ultraselection.** Following Eq. (1), the log likelihood function is given by,

$$\ell(\lambda_s; \mathbb{Y}, \mathbb{P}) = \sum_i Y_i \big[\log(1-\lambda_s) + \log P_i\big] + (1-Y_i)\log\big[1-(1-\lambda_s)P_i\big]$$
$$= R\log(1-\lambda_s) + \sum_{i:Y_i=1}\log P_i + \sum_{i:Y_i=0}\log\big[1-(1-\lambda_s)P_i\big], \quad (2)$$

where $R = \sum_i Y_i$ is the number of rare variants. When the $P_i$ values are small (as is typical), it is possible to obtain a reasonably good closed-form estimator for $\lambda_s$ by making use of the approximation $\log(1-x) \approx -x$. In this case,

$$\ell(\lambda_s; \mathbb{Y}, \mathbb{P}) \approx R\log(1-\lambda_s) + \sum_{i:Y_i=1}\log P_i + \sum_{i:Y_i=0} -(1-\lambda_s)P_i$$
$$= R\log(1-\lambda_s) + \sum_{i:Y_i=1}\log P_i - N\bar{P}'(1-\lambda_s), \quad (3)$$

where $N = \sum_i (1-Y_i)$ is the number of invariant sites and $\bar{P}'$ is the average value of $P_i$ at the invariant sites. It is easy to show that this approximate log likelihood is maximized at,

$$\hat{\lambda}_s = 1 - \frac{R}{N\bar{P}'}. \quad (4)$$

However, this procedure leads to a biased estimator for $\lambda_s$. A correction for the bias leads to the following, intuitively simple, unbiased estimator:

$$\hat{\lambda}_s = 1 - \frac{R}{M\bar{P}}, \quad (5)$$

where $M = N + R$ is the total number of sites and $\bar{P}$ is the average value of $P_i$ at all sites. In other words, $\hat{\lambda}_s$ is given by 1 minus the observed number of rare variants divided by the expected number of rare variants under neutrality, which is simply the total number of sites multiplied by the average rate at which rare variants appear, $\bar{P}$.

**Full allele-specific model**. In practice, we use a model that distinguishes among the alternative alleles at each site and exploits our allele-specific mutation rates. This model behaves similarly to the simpler one described above, but yields slightly more precise estimates in the presence of multi-allelic rare variants.

In the full model, we assume separate indicator variables, $Y_i^{(1)}$, $Y_i^{(2)}$, and $Y_i^{(3)}$, for the three possible allele-specific rare variants at each site, and corresponding allele-specific rates of occurrence, $P_i^{(1)}$, $P_i^{(2)}$, and $P_i^{(3)}$ (which, notably, sum to the quantity previously denoted $P_i$). We further make the assumption that the different rare variants appear independently. Thus, the likelihood function generalizes to (cf. equation (1)),

$$\mathcal{L}(\lambda_s; \mathbb{Y}, \mathbb{P}) = \prod_i \prod_{j=1}^3 \left[ (1-\lambda_s) P_i^{(j)} \right]^{Y_i^{(j)}} \left[ 1 - (1-\lambda_s) P_i^{(j)} \right]^{1-Y_i^{(j)}} \tag{6}$$

where we redefine $\mathbb{Y} = \{Y_i^{(j)}\}$ and $\mathbb{P} = \{P_i^{(j)}\}$ for $j \in \{1, 2, 3\}$. Notice that, when more than one alternative allele is present, $Y_i^{(j)}$ will be 1 for more than one value of $j$.

As for the simplified model above (Eqs. (2)–(5)), the log likelihood can be approximated as,

$$\begin{aligned} \ell(\lambda_s; \mathbb{Y}, \mathbb{P}) &= \sum_i \sum_{j=1}^3 Y_i^{(j)} \left[ \log(1-\lambda_s) + \log P_i^{(j)} \right] + \left(1 - Y_i^{(j)}\right) \log\left[ 1 - (1-\lambda_s) P_i^{(j)} \right] \\ &\approx \log(1-\lambda_s) \left( \sum_i \sum_{j=1}^3 Y_i^{(j)} \right) - (1-\lambda_s) \left( \sum_i \sum_{j=1}^3 \left(1 - Y_i^{(j)}\right) P_i^{(j)} \right) + Z \\ &= R' \log(1-\lambda_s) - N' \bar{Q}' (1-\lambda_s) + Z \end{aligned} \tag{7}$$

where $R' = \sum_i \sum_{j=1}^3 Y_i^{(j)}$ is the total number of rare variants, now allowing for more than one per site; $N' = \sum_i \sum_{j=1}^3 \left(1 - Y_i^{(j)}\right) = 3M - R'$; $\bar{Q}' = \frac{1}{N'} \sum_i \sum_{j=1}^3 \left(1 - Y_i^{(j)}\right) P_i^{(j)}$; and $Z$ is a term that does not depend on $\lambda_s$. This function is maximized at,

$$\hat{\lambda}_s = 1 - \frac{R'}{N' \bar{Q}'}, \tag{8}$$

and a correction for the bias yields an estimator of,

$$\hat{\lambda}_s = 1 - \frac{R'}{(N' + R') \bar{Q}} = 1 - \frac{R'}{M\bar{P}}, \tag{9}$$

where $\bar{Q}$ is the average of all $P_i^{(j)}$ values and we use the facts that $N' + R' = 3M$ and $\bar{P} = 3\bar{Q}$.

When comparing Eqs. (5) and (9), notice that, by construction, $R' \geq R$; thus, the full model will generally lead to slightly smaller estimates of $\lambda_s$ with a difference that reflects the number of multi-allelic rare variants. The two estimators are identical if there are no such sites.

Assuming the $P_i^{(j)}$ values are known, the variance of $\hat{\lambda}_s$ follows from the variance of $R'$, which—because $R'$ is a sum of independent Bernoulli variables—is given by,

$$\begin{aligned} \mathrm{Var}(R') &= \sum_i \sum_{j=1}^3 (1-\lambda_s) P_i^{(j)} \left[ 1 - (1-\lambda_s) P_i^{(j)} \right] \\ &= (1-\lambda_s) M\bar{P} - (1-\lambda_s)^2 T, \end{aligned} \tag{10}$$

where $T = \sum_i \sum_{j=1}^3 \left( P_i^{(j)} \right)^2$. Thus,

$$\begin{aligned} \mathrm{Var}(\hat{\lambda}_s) &= \left( \frac{1}{M\bar{P}} \right)^2 \left[ (1 - \hat{\lambda}_s) M\bar{P} - (1-\hat{\lambda}_s)^2 T \right] \\ &= \frac{1 - \hat{\lambda}_s}{M\bar{P}} - \frac{(1-\hat{\lambda}_s)^2 T}{(M\bar{P})^2} \end{aligned} \tag{11}$$

The standard errors we report for estimates of $\lambda_s$ are obtained by taking the positive square root of this quantity.

When data is simulated under the assumed model, we find that the estimator for $\lambda_s$ (Eqs. (5) and (9)) and the predicted variance (Eq. (11)) agree very well with the truth (Supplementary Fig. 4). Furthermore, if the $P_i^{(j)}$ values are assumed to be random but unbiased, then $\hat{\lambda}_s$ and its standard error have almost no dependency on the variance of $P_i^{(j)}$, at least in the regime of interest. For this reason, we ignore the variance in the mutation-rate estimates when estimating the standard errors for $\lambda_s$.

ExtRaINSIGHT also reports a $p$-value based on a likelihood ratio test of an alternative hypothesis of $\lambda_s \neq 0$ relative to a null hypothesis of $\lambda_s = 0$, assuming twice the log likelihood ratio has an asymptotic $\chi^2$ distribution with one degree of freedom under the null hypothesis.

**Relationship between $s_{het}$ and $\lambda_s$**. When selection against heterozygotes is strong, the equilibrium allele frequency at mutation-selection balance is given by $q = \frac{\mu}{s_{het}}$ (reviewed in ref. [17]). The frequency of mutant alleles in a random sample of $2N$ chromosomes (where $N$ is the number of diploid individuals) will be Poisson-distributed with mean $2N \cdot \frac{\mu}{s_{het}}$ (c.f. ref. [11]), and the expected number of polymorphic sites in a collection of $M$ sites is $E[X] = M(1 - e^{-2N\mu/s_{het}})$. Ignoring common variants for the moment, the same expectation under the ExtRaINSIGHT model is

given by $E[X] = \sum_i (1-\lambda_s) P_i = M(1-\lambda_s)\bar{P}$, where $\bar{P}$ is the mean value of $P_i$ over the sites in question. By setting these quantities equal to one another, we obtain,

$$M(1 - e^{-2N\mu/s_{het}}) = M(1-\lambda_s)\bar{P}$$
$$\frac{2N\mu}{s_{het}} = -\log(1 - (1-\lambda_s)\bar{P}) \approx (1-\lambda_s)\bar{P} \tag{12}$$
$$s_{het} \approx \frac{2N\mu/\bar{P}}{1 - \lambda_s} = \frac{2N/c}{1 - \lambda_s},$$

where $c = \bar{P}/\mu$. With our data, we find that $\bar{P}$ varies little from one set of sites to another, hovering close to $\bar{P} = 0.162$. Assuming $\mu = 1.2 \times 10^{-8}$, we obtain $c = 1.35 \times 10^7$.

This derivation can be adjusted to accommodate common variants (with MAF > 0.001, under our assumptions), but this correction has little effect in practice with our data, because only about 3% of variants are common. Since the relationship is approximate anyway, we use the simpler version above.

It is instructive also to consider the case where $s_{het}$ varies across sites. In this case, if $s_i$ is the selection coefficient against heterozygotes at site $i$ and if each $s_i$ is sufficiently strong for mutation-selection balance to hold, then,

$$M(1-\lambda_s)\bar{P} \approx \sum_i 2N \cdot \frac{\mu}{s_i} = \frac{2MN\mu}{H[s]}$$
$$(1-\lambda_s)\bar{P} \approx \frac{2N\mu}{H[s]}, \tag{13}$$

where $H[s] = \frac{1}{M} \left( \sum_i \frac{1}{s_i} \right)^{-1}$ is the harmonic mean of the $s_i$ values. This relationship is equivalent to the one above but with $H[s]$ in place of $s_{het}$. Therefore, in this case, equation (12) yields an estimator not for the arithmetic mean, but for the harmonic mean of the variable $s_i$ values across sites. It will therefore tend to under-estimate the arithmetic mean in the presence of variable selection. This observation provides an explanation for the downward bias observed in Supplementary Fig. 1.

A further generalization of interest is to assume that a fraction $\pi_0$ of the sites of interest are not under selection at all. In this case, the rare variants will arise as a mixture of sites under selection (and at mutation-selection balance) and sites at which the neutral rate applies. Thus,

$$(1-\lambda_s)\bar{P} \approx (1-\pi_0)\frac{2N\mu}{H[s]} + \pi_0 \bar{P}$$
$$(1 - \lambda_s - \pi_0)\bar{P} \approx (1-\pi_0)\frac{2N\mu}{H[s]} \tag{14}$$
$$H[s] \approx 2N/c \cdot \frac{1-\pi_0}{1 - \lambda_s - \pi_0}.$$

Consequently, if the sites of interest are known to include a component of neutrally evolving sites, and if the fraction $\pi_0$ can be estimated, then a portion of the downward bias in estimation of the selection coefficient can be removed. In particular, the quantity $\rho$ estimated by INSIGHT should function as a fairly good estimate of $1 - \pi_0$. Therefore, if estimates of $\hat{\rho}$ and $\hat{\lambda}_s$ are both available, one can obtain an adjusted estimate of the harmonic mean of $s$ as,

$$H[s] \approx 2N/c \cdot \frac{\hat{\rho}}{\hat{\rho} - \hat{\lambda}_s}. \tag{15}$$

**Application of INSIGHT**. To estimate the total fraction of sites under selection we applied INSIGHT[20,21] in parallel to ExtRaINSIGHT, using the same sets of foreground and background ("neutral") sites. INSIGHT reports a maximum-likelihood estimate of a quantity $\rho$ that measures the fraction of all sites subject to selection on the time scale of the human-chimpanzee divergence (5–7 MY). This quantity includes sites under positive selection as well as those under purifying selection, but for large collections of sites in the human genome the contribution of positive selection is generally negligible (see refs. [21,54]). For efficiency, we used a faster, re-engineered version of INSIGHT, called INSIGHT2, that is mathematically equivalent to the original but performs numerical optimization using the BFGS algorithm rather than expectation maximization[60]. INSIGHT2 is currently only available for the hg19 assembly so we first mapped annotations from hg38 to hg19 using liftOver, ignoring sites outside of regions of one-to-one mapping. We randomly sampled one million sites from larger data sets, to improve efficiency. Notably, INSIGHT makes use of data from Complete Genomics rather than the gnomAD data set for allele-frequency information (see ref. [21]). INSIGHT calculates the standard error of its estimates of $\rho$ by taking the inverse of the corresponding diagonal term of the negative Hessian matrix of the log likelihood function at the MLE.

**Genomic annotations and data processing**. Annotations for CDS, 5′ UTR, 3′ UTR, and introns were defined using the ensembldb Bioconductor package, which interfaces directly with Ensembl. We included only autosomal protein-coding genes. Splice sites were defined as the two nucleotide sites at each of the 5′ and 3′ ends of introns. Within the promotor regions, we used the Ensembl Regulatory Build to locate transcription factor binding sites, which are inferred from experimental data. Flanking regions of TFBS were defined as the 10 bases on either side of

each TFBS. We obtained annotations for lncRNA, snRNA, snoRNA, miRNA also using Ensembl, again restricting them to the autosomes. For all of these annotations, we excluded any regions included in the CDS annotations.

Human accelerated regions (HARs) were obtained from Supplementary Table 1 of ref. [61], a compilation from five previous studies. Ultraconserved noncoding elements (UCNEs) were obtained from UCNEbase[62]. These HARs and UCNEs were defined with respect to hg19, so we mapped them to hg38 using liftOver.

Functional categories were obtained from the Reactome database[31], considering only "top-level" human terms that included at least 100 genes. Tissue specific genes expression data were obtained from Supplementary Table 1 in ref. [63]. Genes were classified as tissue-specific if they had a TS score of greater than three, indicating that they are expressed in that tissue at a level roughly $2^3$ times as high as the average expression level in all other tissues. Note that this definition allows a gene to be "tissue-specific" in more than one tissue. For each category of interest (based on pathway or gene expression), we applied ExtRaINSIGHT to the union of CDS exons of all associated protein-coding gene.

**Simulations**. To test our ability to estimate $s_{het}$ from $\lambda_s$ (as shown in Supplementary Fig. 6), we conducted simulations under a realistic demographic model and various "true" values of $s_{het}$. We then estimated $\lambda_s$ for each data set, converted $\lambda_s$ to $s_{het}$ via equation (12), and compared this estimate to the true value. In each case, we used the simulator developed by Weghorn et al.[18] to generate 100,000 independent nucleotide sites for a population of 71,702 diploid individuals with bottlenecks and growth patterns matching based on a European demographic history. We carried out an initial round of simulations assuming a constant value of $s_{het}$ per simulated data set, with $s_{het}$ ranging from 0.0001 to 0.5, and a second round in which sitewise values of $s_{het}$ were drawn from an exponential distribution with a mean equal to each of the same values. When applying equation (12), we used the mean rate of rare variant occurrence, $\bar{P}$, observed in each simulated data set, which tended to be similar, but not identical, to that from the real data. We assumed a mutation rate of $1.2 \times 10^{-8}$ per generation per site.

In a second series of experiments, we simulated data from DFEs based on real data and evaluated the DFE associated with the "missing" rare variants measured by ExtRaINSIGHT, as well as the quality of the $\lambda_s$ and $s_{het}$ estimators (Supplementary Table 2 and Supplementary Fig. 6). We used four DFEs: (1) one derived from ref. [8] based on data from the 1000 Genomes Project, consisting of a mixture of a point-mass at zero (3.1% weight) and a Gamma distribution with $\alpha = 0.1930$ and $\theta = 0.0168$ ("Kim et al." in Table 2); (2) a version of the same DFE with a larger value of the shape parameter ($\alpha = 0.75$) to better mimic the patterns we observed at 0d sites ("0d CDS" in Table 2); (3) a version with even stronger selection (no point-mass at zero and $\alpha = 0.99$) to mimic the patterns at miRNAs ("miRNA" in Table 2); and (4) a version with substantially weaker selection (a 70% point-mass at zero and $\alpha = 0.45$) to mimic the patterns at TFBSs ("TFBS" in Table 2).

When selecting the DFE from ref. [8], we chose the parameters estimated with a lower mutation rate ($1.5 \times 10^{-8}$), which was close to the one assumed for this study. In addition, when defining DFEs in terms of $s_{het}$, we reduced the reported DFE by a scale factor of $2N_e$ (using the estimated value of $N_e = 12,378$) to account for the population-scaled DFE inferred in ref. [8]. This scaling was accomplished by reducing the value of $\theta$ in the inferred Gamma distribution from 820.6 to 0.0331. Notably, the mean of the DFE estimated for the 1000 Genomes Project data was intermediate between those estimated for the ESP European and LuCAMP data sets in ref. [8].

In each case, we simulated data with the assumed DFE for new mutations, denoted $f(x)$, and then traced the DFE for the rare variants that remained in each data set after selection had been applied, denoted $g(x)$. We then could estimate the DFE for the missing rare variants measured by ExtRaINSIGHT as $h(x) = \frac{1}{\lambda}[f(x) - (1 - \lambda_s)g(x)]$, assuming that the full DFE can be expressed as a mixture of $g(x)$ with weight $1 - \lambda_s$ and $h(x)$ with weight $\lambda_s$. This mixture must also account for common variants, but we omit them because they occur at only a small fraction of sites in our setting.

**Reporting summary**. Further information on research design is available in the Nature Research Reporting Summary linked to this article.

## Data availability
ExtRaINSIGHT and INSIGHT2 scores can be computed for any user-defined set of annotations using the ExtRaINSIGHT web portal at http://compgen.cshl.edu/extrainsight. Auxilary data sources included gnomAD v. 3 (ref. [13]), GENCODE v. 38 (ref. [29]), Reactome[31], the UCSC Genome Browser (hg38)[59], UCNEbase[62], and ref. [61]. Key data files used in our analysis are provided at https://github.com/CshlSiepelLab/extraINSIGHT.

## Code availability
The source code for the ExtRaINSIGHT server and scripts used for data analysis are available at https://github.com/CshlSiepelLab/extraINSIGHT (ref. [64]).

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

## Acknowledgements

We thank Dr. Daniel Balick for providing simulation code from reference [18], and Dr. Shamil Sunyaev for helpful comments. This research was supported by US National Institutes of Health grant R35-GM127070 (to AS) and the Simons Center for Quantitative Biology at Cold Spring Harbor Laboratory. The content is solely the responsibility of the authors and does not necessarily represent the official views of the US National Institutes of Health.

## Author contributions

Y-F.H. proposed the model, implemented an initial version, and carried out an initial analysis of coding and noncoding elements. N.D. re-engineered much of the code and, with help from R.R., developed and released the public server. N.D. also substantially extended the data analysis, introducing the LOEUF scores, reactome analysis and analysis of promoter regions. MRM did the simulation work and carried out the genome-wide accounting of sites. A.S. supervised the research, developed the connections with $s_{het}$ and the analytical estimators for $\lambda_s$ and its variance, and substantially expanded N.D.'s early draft of the manuscript. All authors provided feedback to improve the manuscript, and all authors approved the final version.

## Competing interests

The authors declare no competing interests.
