## [Peer Review File · Nature Communications]

Extreme purifying selection against point mutations in the human genomeREVIEWER COMMENTS

Reviewer #1 (Remarks to the Author):

This manuscript provides a map of ultraselection for the human genome based on the large-scale population sequencing data. This is an interesting and timely work. The manuscript is well written. The authors reiterate the apparent absence of ultraselection at UCEs and report a large fraction of ultraselected sites in ancient miRNAs. The manuscript also places the question of ultraselection within the broader evolutionary theory and underscores a high mutation burden in humans. Still, I have a few questions and comments. I am not sure that some of these are easily addressable but should at least be discussed.

- 1) The most obvious issue with the analysis of ultraselection is the uncertainty of the mutation rate models. For example, this manifests in negative lambda estimates. The authors acknowledge the issue. However, the main result in the manuscript is the quantitative estimates of ultraselection. These estimates are not robust with respect to mutation rate variation. I am not sure this is easily addressable but should be acknowledged with some discussion on the uncertainty of the estimates.
- 2) Faced by the issue of the mutation rate variation, the field of cancer genomics developed a number of overdispersed models that incorporate uncertainty about mutation rate. Human germline genetics did not follow that path and the current work treats mutation rate estimates as exact. It would be helpful to discuss this difference in approaches and a potential of Poisson mixture models.
- 3) The manuscript provides an estimate of the number of "nearly lethal" mutations per generation. I cannot fully understand the "nearly lethal" designation. It seems that their effects may not exceed a few percent of fitness loss. Could the authors clarify that?
- 4) The estimate of 2.21 deleterious mutations per genome (INSIGHT) seems to be slightly lower than the existing estimates based on the analysis of conservation between species. Is the difference because of the accounting for the mutation rate variation or something else? The authors do discuss that their splice site estimate seems a little lower than previous estimates and their estimates of s_{het} for missense mutations seems slightly higher. This is a good discussion.
- 5) It might be important for some in the field to reiterate that the estimate of 2.21 is not consistent with the population survival under a purely multiplicative model for independent mutation effects (the classic mutation load argument).

Reviewer #2 (Remarks to the Author):

The authors introduce a new statistical model to infer the strength of negative selection using large genomic datasets. The rationale behind their method is that with a large enough sample (in the specific case of the analyses they conduct in this manuscript, ~70,000 genome sequences) we should expect to have a representative collection of sites that are (in)tolerant to mutations. Then, by contrasting the number of variants in a set of putatively constrained sites with a matching set of putatively neutral sites, the observed depletion of variation in the former group is a consequence of negative selection. The main result of the paper is a set of equations that first estimates such depletion in variation relative to the neutral expectation (accounting for mutation rate variation along the genome) and then relates this measure to the strength of negative selection against heterozygous genotypes, under the assumption of mutation-selection balance.

The overall approach seems clever and innovative. While this model represents an interesting way to look for the signature of negative selection using the large datasets of whole-genome sequences that are becoming available, there are a number of limitations of the method that were not fully examined, addressed, or articulated in the current version of the manuscript. Thus, we do not believe that the main conclusions of the paper, that "0.3–0.5% of the human genome is ultraselected", "that 22% of 0-fold sites are ultraselected, meaning that any mutation at these sites would be nearly lethal" and

“that $s_{het} = 0.014$ ” are supported. These specific concerns are outlined below:

Major comments

1) The work presented by the authors leans heavily on assumptions of mutation-selection balance. It is still unclear that what extent the mutation-selection balance holds under scenarios of complex demographic history like in humans. Evidence more reliable than population genetics shows that very recent human history (which is the time-scale where mutation-selection balance would operate) involves complicated migrations, range expansions and super exponential growth. Nevertheless, the authors justify the applicability of the mutation-selection balance using previous work from (Cassa et al. 2017) (line 56: “[...] this relationship holds quite well for pLoF variants in the ExAC exome data [12] down to $s_{het} \sim 0.01$ ”). However, the adequacy of the results from Cassa et al have been contested by (Charlesworth and Hill 2019), who argued that fluctuations around the expected equilibrium frequency caused by genetic drift cannot be so easily neglected. In response, (Weghorn et al. 2019) have published a new study where they test the applicability of mutation-selection balance to infer s_{het} , concluding that “However, variance indeed diverged from the deterministic approximation for genes under weaker selection ($s_{het} \leq 0.02$)”. Crucially, this lower limit for accurate inference under mutation-selection balance lies above the vast majority of selection coefficients inferred by authors in the present work.

2) The statistical model assumes that among candidate sites putatively under strong negative selection, non-segregating sites are constrained and, conversely, segregating sites are not. In other words, their method infers selection based on presence/absence states and ignores allele frequency. To try and assess the validity of these assumptions, the authors perform a simulation study showing that their method has remarkable accuracy for very strong selection ($s_{het} > 0.1$, Figure S3), but performs poorly when $s_{het} \leq 0.03$ (and not, in fact, $s_{het} \leq 0.013$ as the authors claim on line 121). Indeed, according to their own simulations, the value of $s_{het} \sim 0.013$ (which already represents strong selection) seems to be the floor of what their method outputs, regardless of how weak negative selection actually is in the simulations. Unfortunately, the results they find in real data fall nearly always in such region of poor performance. This means that most of the biological conclusions they draw are statistically unreliable. The authors claim that estimates in the region of $s_{het} \sim 0.013$ are “inflated but approximately useful”, but this claim has no objective support.

3) This is further problematic in real data (where the selection coefficient is not uniform across sites) since their estimate of s_{het} assumes that mutation-selection balance holds for all candidate sites, including those in the lower tail of the DFE (equation 9 in Methods). Consequently, since such weakly deleterious variants deviate even more strongly from mutation-selection balance expectations, and given that their method estimates s_{het} with a high floor (Figure S3), the final estimate may be substantially inflated relative to the true (unknown) harmonic mean of the DFE of candidate sites.

4) Despite evidence that the results are inaccurate for $s_{het} \leq 0.03$, the authors ignore such limitations and proceed to apply it to different sets of candidate sites. The series of results presented sometimes involves strong claims. Here are some specific examples:

a. Line 133: “we can interpret these estimates as indicating that 22% of 0-fold sites are ultraselected, meaning that any mutation at these sites would be nearly lethal [...]”. See comments below for why we believe the terms “ultraselection” and “nearly lethal” are unwarranted in the context of this article.

b. Lines 340-360 discuss about the discordance between the authors’ results and previous estimates using the site frequency spectrum. Although the authors en passant acknowledge the low accuracy of their s_{het} estimates, once again they neglect it in order to reach a provocative statement at end the paragraph: “It therefore seems plausible that our fourfold higher estimate of $s_{het} \sim 0.014$ is closer to the true mean value than these SFS-based estimates.” The proposal to replace existing SFS-based estimates of the DFE seems unfounded in view of the low accuracy of their method around s_{het}

~ 0.014 and non-uniformity of selection strength across sites. Thus, at a minimum, this claim should be toned-down and more nuance should be provided. The authors also seem to compare the (arithmetic) mean selection coefficient of the DFE inferred by (Kim, Huber, and Lohmueller 2017) with their own estimate of s_{het} , which reflects the harmonic mean of the DFE instead (Figure S3 and Methods section, line ~ 500). This comparison is misleading.

c. We note that one very interesting result presented by the authors is the inference of s_{het} in ultraconserved noncoding elements, where they report surprisingly weak selection, corroborating previous studies (lines 310-334). The precise evolutionary mechanism invoked as an explanation (weak selection in short time-scales preventing fixation in long time-scales, at the phylogenetic level), however, needs to be refined. Under what conditions should we expect that genomic regions with increased weakly deleterious diversity will not end up with increased fixation rate of weakly deleterious mutations (thus being visible as divergence)?

5) The authors present a new DFE (in Figure S5) that they suggest can match the mean s_{het} values. Does this DFE actually fit the SFS of nonsynonymous variants for some human genetic variation dataset? Additionally, there seems to be a mis-match between the color scheme used in the figure, the legend, and the caption. The distinctions between $f(x)$, $g(x)$, and $h(x)$ were not always clear.

6) The authors report that λ_S takes on negative values for some sets of candidate sites, which is biologically implausible following their own assumptions. They acknowledge that this may be due to mis-specification of the mutational model (line 107). It seems possible that the same issue may arise in other sets of candidate sites as well, even if not leading to negative λ_S . Thus, to what extent could mis-specification of the mutational model be driving the conclusion regarding the presence of "ultraselection"? The authors should conduct simulations to assess the consequences of mis-specification of the mutational model on downstream inferences. Further demonstration of the adequacy of the fit of the mutational model is also required.

7) In addition to addressing the points above, we suggest the following modifications and additional analyses:

a. The term "ultraselection" should be replaced since there is neither need nor justification for it, based on the results presented. For example, (Cassa et al. 2017) found stronger negative selection against protein-truncating variants and were able to describe their results using standard terminology.

b. Likewise, there is no reason to re-define "lethal" and "nearly lethal" mutations based on expected sojourning time. "Lethal" has a clear meaning, and there have been many rigorous studies analyzing the population genetics of homozygous and overdominant lethal mutations, i.e., those that cause death when their effects are fully manifested, e.g. see (Ballinger and Noor 2018) and references therein.

c. Figures S2 and S3 should be moved from supplemental to part of the main article since they display information that is vital for the interpretation of the results. They can be presented as panels in the same figure. If the total number of display items would become an issue as a consequence, Figures 1-4 can safely be pooled together in some combination, since they share the same overall structure.

d. The results are presented (in the figures and throughout the text) mostly in terms of the depletion of variants relative to neutrality, λ_S , which makes them hard to interpret. While λ_S is a parameter of the statistical model the authors describe, in population genetics parlance λ_S is more closely related to a summary statistic. In other words, in order to be meaningful, the depletion of genetic variation in candidate sites must be explained by evolutionary processes, in this case either by mutation rate variation (which the authors control for) or natural selection, represented by s_{het} , the parameter of interest in this paper. We therefore suggest that figures and text switch from λ_S to s_{het} values,

whenever possible.

References

Ballinger, Mallory A., and Mohamed A. F. Noor. 2018. "Are Lethal Alleles Too Abundant in Humans?" *Trends in Genetics: TIG* 34 (2): 87–89. <https://doi.org/10.1016/j.tig.2017.12.013>.

Cassa, Christopher A., Donate Weghorn, Daniel J. Balick, Daniel M. Jordan, David Nusinow, Kaitlin E. Samocha, Anne O'Donnell-Luria, et al. 2017. "Estimating the Selective Effects of Heterozygous Protein-Truncating Variants from Human Exome Data." *Nature Genetics* 49 (5): 806–10. <https://doi.org/10.1038/ng.3831>.

Charlesworth, Brian, and William G. Hill. 2019. "Selective Effects of Heterozygous Protein-Truncating Variants." *Nature Genetics* 51 (1): 2–2. <https://doi.org/10.1038/s41588-018-0291-9>.

Kim, Bernard Y., Christian D. Huber, and Kirk E. Lohmueller. 2017. "Inference of the Distribution of Selection Coefficients for New Nonsynonymous Mutations Using Large Samples." *Genetics* 206 (1): 345–61. <https://doi.org/10.1534/genetics.116.197145>.

Weghorn, Donate, Daniel J Balick, Christopher Cassa, Jack A Kosmicki, Mark J Daly, David R Beier, and Shamil R Sunyaev. 2019. "Applicability of the Mutation–Selection Balance Model to Population Genetics of Heterozygous Protein-Truncating Variants in Humans." *Molecular Biology and Evolution* 36 (8): 1701–10. <https://doi.org/10.1093/molbev/msz092>.

We thank the two reviewers for their reading of our manuscript, and for their thoughtful and constructive suggestions for improvement. In response, we have made a number of key changes to our manuscript (highlighted in the margins) and carried out additional validation of our findings. Below we provide detailed point-by-point responses to the comments and concerns raised by the reviewers (in blue).

REVIEWER COMMENTS

Reviewer #1 (Remarks to the Author):

This manuscript provides a map of ultraselection for the human genome based on the large-scale population sequencing data. This is an interesting and timely work. The manuscript is well written. The authors reiterate the apparent absence of ultraselection at UCEs and report a large fraction of ultraselected sites in ancient miRNAs. The manuscript also places the question of ultraselection within the broader evolutionary theory and underscores a high mutation burden in humans. Still, I have a few questions and comments. I am not sure that some of these are easily addressable but should at least be discussed.

We are pleased to find that the reviewer finds the work interesting, timely, and well written, and we appreciate the constructive suggestions for improvement.

1) The most obvious issue with the analysis of ultraselection is the uncertainty of the mutation rate models. For example, this manifests in negative lambda estimates. The authors acknowledge the issue. However, the main result in the manuscript is the quantitative estimates of ultraselection. These estimates are not robust with respect to mutation rate variation. I am not sure this is easily addressable but should be acknowledged with some discussion on the uncertainty of the estimates.

We agree that the mutation model is a critical feature of the analysis and the most difficult part to get right. In our revised manuscript, we have expanded our treatment of this issue in four ways.

First, we expand our comparison of real and predicted counts of rare variants to consider the variance as well as the mean of the two distributions, by examining empirical histograms in comparison to ones obtained by sampling under our model (Supplementary Figs. S1 & S2). We find that, in addition to providing a good match to the empirical means, our model provides a reasonably good match to the empirical variance. We do observe an excess of rare variants in some classes of sites, particularly in promoter regions and repetitive sequences (Supplementary Fig. S2). Importantly, however, this bias always seems to occur in the direction of too many, rather than too few, observed rare variants relative to the model—which will tend to make our estimates of λ_s conservative (i.e., too low rather than too high).

Second, in a new analysis, we compare the predictions of our mutation model with corresponding rates of *de novo* mutations from denovo-db (Supplemental Fig. S3). This data

set is somewhat sparse (we were able to map about 175,000 mutations to our data) but we believe it is worth showing because it should be even less influenced by natural selection than the rare variants from gnomAD. We find that our model-based predictions match the relative rates at which these rare variants occur reasonably well across a large range of P_i values (though data is sparse at the high end).

Third, we derive new analytical expressions for both the MLE of λ_s and its variance (previously we had computed the MLE numerically and approximated the variance) and we now show by simulation that the variance of estimates of λ_s has essentially no dependency on the variance of the individual P_i estimates (see Supplementary Fig. S4). Therefore, unless the P_i 's are systematically biased—which from the analysis above does not seem to be true in general (with a few rare exceptions)—we do not expect them to expand the range of plausible values of λ_s .

Finally, using our analytical estimates of variance, we now propagate uncertainty in estimates of λ_s through our genome-wide analyses. Based on our new analytical methods, we compute standard errors for all estimates of λ_s (now shown in Tables 1 and S1). We then aggregate these values in accounting for uncertainty in our estimates of the fraction of the genome that is ultraselected and the total number of strongly deleterious mutations. As it turns out, these genome-wide standard errors are quite small—almost certainly negligible in comparison to our conservative correction for model misspecification and uncertainty about the genome-wide mutation rate—but nonetheless we report them throughout.

We now summarize these new analyses in a series of Supplementary Figures (S1–S4) and briefly describe them in the main text when introducing the mutation model. In addition, we have added a paragraph to the Discussion (second paragraph) to address the challenges of fully accounting for variation in mutation rate and summarize our attempts to validate our model.

2) Faced by the issue of the mutation rate variation, the field of cancer genomics developed a number of overdispersed models that incorporate uncertainty about mutation rate. Human germline genetics did not follow that path and the current work treats mutation rate estimates as exact. It would be helpful to discuss this difference in approaches and a potential of Poisson mixture models.

As described above, we believe we have now found a reasonable way to accommodate uncertainty in mutation rate within our original framework, but we appreciate the contrast with the models used in cancer genomics and now highlight this point in the discussion.

3) The manuscript provides an estimate of the number of “nearly lethal” mutations per generation. I cannot fully understand the “nearly lethal” designation. It seems that their effects may not exceed a few percent of fitness loss. Could the authors clarify that?

We had been aiming for an evocative description of the strong levels of purifying selection indicated by the depletion of rare variants, but on reflection, we concede that “nearly lethal” is a

confusing and potentially misleading term. We have replaced it throughout with the more anodyne term “strongly deleterious”.

4) The estimate of 2.21 deleterious mutations per genome (INSIGHT) seems to be slightly lower than the existing estimates based on the analysis of conservation between species. Is the difference because of the accounting for the mutation rate variation or something else? The authors do discuss that their splice site estimate seems a little lower than previous estimates and their estimates of s_{het} for missense mutations seems slightly higher. This is a good discussion.

The estimate from INSIGHT is conservative owing to the correction for model misspecification (i.e., the subtraction of the estimates for nonconserved noncoding regions). We now call attention to this point in the Discussion, and cite some comparison points from key papers that have addressed this question. We find, overall, that our INSIGHT-based estimate is reasonably compatible with previous estimates, especially considering that it is designed to err on the side of being conservative.

5) It might be important for some in the field to reiterate that the estimate of 2.21 is not consistent with the population survival under a purely multiplicative model for independent mutation effects (the classic mutation load argument).

We now remind the reader of this point in the Discussion (with key references).

Reviewer #2 (Remarks to the Author):

The authors introduce a new statistical model to infer the strength of negative selection using large genomic datasets. The rationale behind their method is that with a large enough sample (in the specific case of the analyses they conduct in this manuscript, ~70,000 genome sequences) we should expect to have a representative collection of sites that are (in)tolerant to mutations. Then, by contrasting the number of variants in a set of putatively constrained sites with a matching set of putatively neutral sites, the observed depletion of variation in the former group is a consequence of negative selection. The main result of the paper is a set of equations that first estimates such depletion in variation relative to the neutral expectation (accounting for mutation rate variation along the genome) and then relates this measure to the strength of negative selection against heterozygous genotypes, under the assumption of mutation-selection balance.

The overall approach seems clever and innovative. While this model represents an interesting way to look for the signature of negative selection using the large datasets of whole-genome sequences that are becoming available, there are a number of limitations of the method that were not fully examined, addressed, or articulated in the current version of the manuscript. Thus, we do not believe that the main conclusions of the paper, that “0.3–0.5% of the human genome is ultraselected”, “that 22% of 0-fold sites are ultraselected, meaning that any mutation

at these sites would be nearly lethal” and “that $s_{het} = 0.014$ ” are supported. These specific concerns are outlined below:

Major comments

1) The work presented by the authors leans heavily on assumptions of mutation-selection balance. It is still unclear that what extent the mutation-selection balance holds under scenarios of complex demographic history like in humans. Evidence more reliable than population genetics shows that very recent human history (which is the time-scale where mutation-selection balance would operate) involves complicated migrations, range expansions and super exponential growth. Nevertheless, the authors justify the applicability of the mutation-selection balance using previous work from (Cassa et al. 2017) (line 56: “[...] this relationship holds quite well for pLoF variants in the ExAC exome data [12] down to $s_{het} \sim 0.01$ ”). However, the adequacy of the results from Cassa et al have been contested by (Charlesworth and Hill 2019), who argued that fluctuations around the expected equilibrium frequency caused by genetic drift cannot be so easily neglected. In response, (Weghorn et al. 2019) have published a new study where they test the applicability of mutation-selection balance to infer s_{het} , concluding that “However, variance indeed diverged from the deterministic approximation for genes under weaker selection ($s_{het} \leq 0.02$)”. Crucially, this lower limit for accurate inference under mutation-selection balance lies above the vast majority of selection coefficients inferred by authors in the present work.

We fear we may have inadvertently created a mistaken impression that our entire analysis “leans heavily” on the assumptions of mutation-selection balance. In fact, what we regard as the main findings of the paper—the measures of ultraselection for particular classes of genomic sites and for the genome as a whole—do not depend at all on these assumptions. It is only the analytical estimation of s_{het} from λ_s —which we regard more as a device for gaining intuition about λ_s than a key result—that depends on mutation-selection balance. In our revision, we have tried to make this point much clearer. As described below, we have also reduced the emphasis on the s_{het} estimates in the early parts of the manuscript. Finally, we now cite the commentary by Charlesworth and Hill, together with the works by Cassa et al. and Weghorn et al., when introducing the mutation-selection balance ideas.

2) The statistical model assumes that among candidate sites putatively under strong negative selection, non-segregating sites are constrained and, conversely, segregating sites are not. In other words, their method infers selection based on presence/absence states and ignores allele frequency. To try and assess the validity of these assumptions, the authors perform a simulation study showing that their method has remarkable accuracy for very strong selection ($s_{het} > 0.1$, Figure S3), but performs poorly when $s_{het} \leq 0.03$ (and not, in fact, $s_{het} \leq 0.013$ as the authors claim on line 121). Indeed, according to their own simulations, the value of $s_{het} \sim 0.013$ (which already represents strong selection) seems to be the floor of what their method outputs, regardless of how weak negative selection actually is in the simulations. Unfortunately, the results they find in real data fall nearly always in such region of poor performance. This means that most of the biological conclusions they draw are statistically

unreliable. The authors claim that estimates in the region of $s_{\text{het}} \sim 0.013$ are “inflated but approximately useful”, but this claim has no objective support.

We acknowledge that the threshold for applicability of the s_{het} estimator appeared to be somewhat higher than desired, and that we were overly aggressive about the range in which we applied it. After further consideration, we have decided to substantially scale back our emphasis on s_{het} throughout the manuscript. We now are clear from the outset that λ_s is the main focus of our analysis and that the estimates of s_{het} depend on the assumptions of mutation-selection balance and are only reliable for quite large values of λ_s . We now apply them only when $\lambda_s > 0.45$, rather than when $\lambda_s > 0.18$, as in the previous manuscript. We have moved the introduction of the key s_{het} equation from the results section to the methods. We have also revised our discussion extensively to reduce the emphasis on s_{het} estimates.

At the same time, it is worth noting that we revisited the details of our simulations and realized that we had made an error that caused our previous results to be overly pessimistic. Briefly, when we initially simulated data under the Tennesen et al. model and a realistic choice of the mutation rate, we observed substantially fewer (by about two-fold) rare variants than are evident in the real data. To compensate, we had naively increased the mutation rate from $1.2e-8$ to $2.2e-8$ mutations per generation per site. But we now find that if we instead increase the rate of recent population growth in the Tennesen et al. model from 1.9 to 4.1% (in qualitative agreement with Keinan & Clark, 2012) and keep the mutation rate at $1.2e-8$ mutations per generation per site, we can match the frequency of rare variants in the real data without an excess of common variants. In this case, the s_{het} estimator turns out to behave considerably better, showing excellent agreement with the truth down to about $s_{\text{het}}=0.03$ and acceptable agreement down to $s_{\text{het}}=0.02$ (roughly equivalent to the previous performance at $s_{\text{het}}=0.04$; see new Supplemental Fig. S5). With these improved results, we now make use of the estimator in cases where the estimated $s_{\text{het}} > 0.02$, which corresponds to $\lambda_s > 0.45$. We eliminated use of the intermediate (light gray) region where, as the reviewer noted, we had awkwardly regarded the estimator as “inflated but approximately useful”

3) This is further problematic in real data (where the selection coefficient is not uniform across sites) since their estimate of s_{het} assumes that mutation-selection balance holds for all candidate sites, including those in the lower tail of the DFE (equation 9 in Methods). Consequently, since such weakly deleterious variants deviate even more strongly from mutation-selection balance expectations, and given that their method estimates s_{het} with a high floor (Figure S3), the final estimate may be substantially inflated relative to the true (unknown) harmonic mean of the DFE of candidate sites.

As described above, we have substantially reduced our emphasis on estimates of s_{het} in the analysis of real data, and further acknowledged the limitations of the estimator in both the results and discussion section.

4) Despite evidence that the results are inaccurate for $s_{het} \leq 0.03$, the authors ignore such limitations and proceed to apply it to different sets of candidate sites. The series of results presented sometimes involves strong claims. Here are some specific examples:

a. Line 133: “we can interpret these estimates as indicating that 22% of 0-fold sites are ultraselected, meaning that any mutation at these sites would be nearly lethal [...]”. See comments below for why we believe the terms “ultraselection” and “nearly lethal” are unwarranted in the context of this article.

As discussed in the response to reviewer #1, we agree that the term “nearly lethal” is unsuitable and have eliminated it. However, we feel that the term “ultraselection” is useful as long as it is clearly defined (see below).

b. Lines 340-360 discuss about the discordance between the authors’ results and previous estimates using the site frequency spectrum. Although the authors en passant acknowledge the low accuracy of their s_{het} estimates, once again they neglect it in order to reach a provocative statement at end the paragraph: “It therefore seems plausible that our fourfold higher estimate of $s_{het} \sim 0.014$ is closer to the true mean value than these SFS-based estimates.” The proposal to replace existing SFS-based estimates of the DFE seems unfounded in view of the low accuracy of their method around $s_{het} \sim 0.014$ and non-uniformity of selection strength across sites. Thus, at a minimum, this claim should be toned-down and more nuance should be provided. The authors also seem to compare the (arithmetic) mean selection coefficient of the DFE inferred by (Kim, Huber, and Lohmueller 2017) with their own estimate of s_{het} , which reflects the harmonic mean of the DFE instead (Figure S3 and Methods section, line ~500). This comparison is misleading.

In our view, the primary observation described in this paragraph is that data simulated under the DFE inferred by Kim et al. results in a substantially lower estimate of λ_s (previously by five-fold, now by about three-fold with our new demographic model) than we infer from real data. This observation does not depend on our estimates of s_{het} nor on the assumption of mutation-selection balance; hence we stand by our claim that “the patterns of rare variants present in the deeply sequenced gnomAD data set do not seem to be consistent with the DFEs inferred from smaller data sets”. However, in response to the reviewer’s comments, we have rewritten this paragraph to focus it more clearly on λ_s and to be clear that we cannot estimate s_{het} with any accuracy in coding regions.

c. We note that one very interesting result presented by the authors is the inference of s_{het} in ultraconserved noncoding elements, where they report surprisingly weak selection, corroborating previous studies (lines 310-334). The precise evolutionary mechanism invoked as an explanation (weak selection in short time-scales preventing fixation in long time-scales, at the phylogenetic level), however, needs to be refined. Under what conditions should we expect that genomic regions with increased weakly deleterious diversity will not end up with increased fixation rate of weakly deleterious mutations (thus being visible as divergence)?

To address this question, we now perform a simulation experiment to identify a range of s_{het} values that produce qualitatively similar patterns to what we observe in UCNEs, in terms of both ultraselection and cross-species conservation (Supplementary Fig. S8). We find that values of s_{het} between approximately 0.003 and 0.005 lead to near perfect cross-species conservation but only modest values of λ_s (roughly 0.10–0.15). We now mention this result in the relevant paragraph of the Discussion.

5) The authors present a new DFE (in Figure S5) that they suggest can match the mean s_{het} values. Does this DFE actually fit the SFS of nonsynonymous variants for some human genetic variation dataset? Additionally, there seems to be a mis-match between the color scheme used in in the figure, the legend, and the caption. The distinctions between $f(x)$, $g(x)$, and $h(x)$ were not always clear.

The purpose of this experiment is to evaluate the DFEs of the “missing” rare variants—i.e., the rare variants whose depletion is measured by the λ_s parameter—for a variety of plausible DFEs. Toward this end, we considered the DFE estimated by Kim et al. as well as three variants of that DFE designed to approximately match our λ_s estimates for 0d sites in coding regions, ancient miRNAs, and TFBSs. Our concern was that, under certain conditions, λ_s might be driven by sites under weaker selection rather than sites at which mutations are strongly deleterious. We find, to the contrary, that the “missing” rare variants always occur at sites under fairly strong selection (with mean values of $s_{\text{het}} \approx 0.02$), for a range of initial DFEs. Thus, λ_s does indeed seem to be measuring quite strong purifying selection. We have tried to clarify these points further in the revised manuscript. We have also checked the colors and legends and verified that they are all correct and in agreement with one another.

6) The authors report that λ_S takes on negative values for some sets of candidate sites, which is biologically implausible following their own assumptions. They acknowledge that this may be due to mis-specification of the mutational model (line 107). It seems possible that the same issue may arise in other sets of candidate sites as well, even if not leading to negative λ_S . Thus, to what extent could mis-specification of the mutational model be driving the conclusion regarding the presence of “ultraselection”? The authors should conduct simulations to assess the consequences of mis-specification of the mutational model on downstream inferences. Further demonstration of the adequacy of the fit of the mutational model is also required.

As detailed in the response to reviewer #1, we agree that misspecification of the mutation model is a critical issue to address, and we have now added several analyses both to validate the model and to better account for uncertainty in the λ_s estimates.

7) In addition to addressing the points above, we suggest the following modifications and additional analyses:

a. The term “ultraselection” should be replaced since there is neither need nor justification for it, based on the results presented. For example, (Cassa et al. 2017) found stronger negative

selection against protein-truncating variants and were able to describe their results using standard terminology.

We believe that we introduce the term “ultraselection” fairly clearly and forthrightly as a form of shorthand that enables concise descriptions of the phenomena we observe throughout the article. We have found that this term resonates with other readers and that they do not find it confusing or misleading. Therefore, we wish to retain it, with the editor’s permission.

b. Likewise, there is no reason to re-define “lethal” and “nearly lethal” mutations based on expected sojourning time. “Lethal” has a clear meaning, and there have been many rigorous studies analyzing the population genetics of homozygous and overdominant lethal mutations, i.e., those that cause death when their effects are fully manifested, e.g. see (Ballinger and Noor 2018) and references therein.

As noted in the response to reviewer #1, we have replaced the term “nearly lethal” with “strongly deleterious” throughout the manuscript.

c. Figures S2 and S3 should be moved from supplemental to part of the main article since they display information that is vital for the interpretation of the results. They can be presented as panels in the same figure. If the total number of display items would become an issue as a consequence, Figures 1-4 can safely be pooled together in some combination, since they share the same overall structure.

Given that we now downplay the s_{het} estimates and focus the article more clearly on λ_s , we have decided to retain these as supplementary figures.

d. The results are presented (in the figures and throughout the text) mostly in terms of the depletion of variants relative to neutrality, λ_S , which makes them hard to interpret. While λ_S is a parameter of the statistical model the authors describe, in population genetics parlance λ_S is more closely related to a summary statistic. In other words, in order to be meaningful, the depletion of genetic variation in candidate sites must be explained by evolutionary processes, in this case either by mutation rate variation (which the authors control for) or natural selection, represented by s_{het} , the parameter of interest in this paper. We therefore suggest that figures and text switch from λ_S to s_{het} values, whenever possible.

For the reasons described above, we believe it is better to retain the original convention of describing the results primarily in terms of λ_s and providing information about s_{het} only in certain cases (when λ_s is sufficiently large) as a way of interpreting the λ_s estimates.

References

Ballinger, Mallory A., and Mohamed A. F. Noor. 2018. “Are Lethal Alleles Too Abundant in Humans?” *Trends in Genetics*: TIG 34 (2): 87–89. <https://doi.org/10.1016/j.tig.2017.12.013>.

Cassa, Christopher A., Donate Weghorn, Daniel J. Balick, Daniel M. Jordan, David Nusinow, Kaitlin E. Samocha, Anne O'Donnell-Luria, et al. 2017. "Estimating the Selective Effects of Heterozygous Protein-Truncating Variants from Human Exome Data." *Nature Genetics* 49 (5): 806–10. <https://doi.org/10.1038/ng.3831>.

Charlesworth, Brian, and William G. Hill. 2019. "Selective Effects of Heterozygous Protein-Truncating Variants." *Nature Genetics* 51 (1): 2–2. <https://doi.org/10.1038/s41588-018-0291-9>.

Kim, Bernard Y., Christian D. Huber, and Kirk E. Lohmueller. 2017. "Inference of the Distribution of Selection Coefficients for New Nonsynonymous Mutations Using Large Samples." *Genetics* 206 (1): 345–61. <https://doi.org/10.1534/genetics.116.197145>.

Weghorn, Donate, Daniel J Balick, Christopher Cassa, Jack A Kosmicki, Mark J Daly, David R Beier, and Shamil R Sunyaev. 2019. "Applicability of the Mutation–Selection Balance Model to Population Genetics of Heterozygous Protein-Truncating Variants in Humans." *Molecular Biology and Evolution* 36 (8): 1701–10. <https://doi.org/10.1093/molbev/msz092>.

REVIEWERS' COMMENTS

Reviewer #1 (Remarks to the Author):

The revised manuscript addresses two previous concerns by the reviewers. First, it investigates accuracy (and occasional misspecification) of mutation rate estimates. The issue of misspecification within some interesting functional categories has not been fully resolved, but the manuscript acknowledges this complication and provides an extensive discussion of the issue including informative supplementary figures. Given the complexity of the issue, I find that this revision is sufficient. Second, the authors investigated the relationship between λ and s_{het} . It was shown that the theoretically predicted relationship holds in the region where mutation-selection balance is applicable to human data (around $s_{het} > 0.02$) in agreement with Weghorn et al., MBE 2019. This is a welcome addition. It is interesting that the authors compare λ estimates per gene to LOEUF estimates rather than s_{het} estimates. However, this choice is probably justified because the LOEUF estimates are based on number of segregating sites similarly to λ estimates in contrast to published s_{het} estimates that are based on the total allele frequency of PTVs. Last, I find that a new deeper population genetics discussion improves the manuscript.

I have only one minor comment. The results are presented primarily as a set of numbers and fractions of ultra-selected sites and the mutation rate in the strongly deleterious class. For an inattentive reader, these numbers look like facts about genetics, while in fact they are very much dependent on voluntarily selected thresholds. It would be great to reflect this in the text.

Reviewer #2 (Remarks to the Author):

Overall, I believe that the authors performed an excellent revision of the manuscript, which now reads much better and accurately reflects their results. The method is interesting, innovative, and presents a clever way to exploit large datasets in population genomics. Importantly, the edits do not reduce the impact of the work. If anything, they make the results more believable. Taking a fresh look at it, the switch in focus from s_{het} to λ_s works well. While it reduces interpretability of the results a little (because λ_s is not an evolutionary parameter), it represents a good compromise and the authors are still able to draw interesting biological conclusions.

I only have the following remaining comments:

Line 355-360: I like the new results on the parameters for selection on UCNEs. I think this is a nice contribution to the paper. In addition to the number of sites under selection, might it also be the case that another unusual thing about UCNEs is that they are under selection in many species? This is reflected in the reduced divergence across species. Maybe consider adding a sentence or two about this?

Lines 370: When analyzing the Kim et al. DFE, the authors write, "For example, the best-fitting such model in a representative recent study by Kim et al. [8], based on a fairly large sample size (432 Europeans from the 1000 Genomes Project), implied a mean selection coefficient against amino-acid replacements of $s = 0.007$, corresponding to only $s_{het} = 1/2s = 0.0035$, since this model assumed additivity." I think there is a factor of 2 error here. The Kim et al. DFE, as well as FitDadi and the previous PRF models (Williamson et al 2004 & Williamson et al. 2005; Boyko et al. 2008) parameterize the fitnesses as 1, $1+s$, and $1+2s$. Thus, the value of s that is inferred from these SFS-based methods corresponds to the fitness effect in the heterozygote, not the homozygote. Thus, the authors seem to be incorrectly dividing the scale parameter of the gamma DFE from Kim et al. by 2 in their analysis. This would result in them simulating less selection than inferred by Kim et al., perhaps accounting for some of the discrepancy in the models. The authors should investigate this more and adjust the conclusions about the differences in DFEs appropriately.

Table S2: Examination of the parameters here support my point above—the scale parameter for the gamma DFE in terms of the heterozygous fitness effect should be 0.03314752. Also, I believe there is a typo here for the shape parameter. It should be 0.199 and not 0.1930 as reported in the table.

REVIEWERS' COMMENTS

Reviewer #1 (Remarks to the Author):

The revised manuscript addresses two previous concerns by the reviewers. First, it investigates accuracy (and occasional misspecification) of mutation rate estimates. The issue of misspecification within some interesting functional categories has not been fully resolved, but the manuscript acknowledges this complication and provides an extensive discussion of the issue including informative supplementary figures. Given the complexity of the issue, I find that this revision is sufficient. Second, the authors investigated the relationship between λ and s_{het} . It was shown that the theoretically predicted relationship holds in the region where mutation-selection balance is applicable to human data (around $s_{het} > 0.02$) in agreement with Weghorn et al., MBE 2019. This is a welcome addition. It is interesting that the authors compare λ estimates per gene to LOEUF estimates rather than s_{het} estimates. However, this choice is probably justified because the LOEUF estimates are based on number of segregating sites similarly to λ estimates in contrast to published s_{het} estimates that are based on the total allele frequency of PTVs. Last, I find that a new deeper population genetics discussion improves the manuscript.

We thank the reviewer for these comments and are glad to see that our revised manuscript is acceptable.

I have only one minor comment. The results are presented primarily as a set of numbers and fractions of ultra-selected sites and the mutation rate in the strongly deleterious class. For an inattentive reader, these numbers look like facts about genetics, while in fact they are very much dependent on voluntarily selected thresholds. It would be great to reflect this in the text.

We have added a sentence to the discussion to emphasize the point that the quantitative results are inherently dependent on a variety of subjective decisions.

Reviewer #2 (Remarks to the Author):

Overall, I believe that the authors performed an excellent revision of the manuscript, which now reads much better and accurately reflects their results. The method is interesting, innovative, and presents a clever way to exploit large datasets in population genomics. Importantly, the edits do not reduce the impact of the work. If anything, they make the results more believable. Taking a fresh look at it, the switch in focus from s_{het} to λ_s works well. While it reduces interpretability of the results a little (because λ_s is not an evolutionary parameter), it represents a good compromise and the authors are still able to draw interesting biological conclusions.

We agree that the revised manuscript is improved and thank the reviewer for the constructive comments regarding s_{het} , which encouraged us to fundamentally rethink this aspect of our presentation.

I only have the following remaining comments:

Line 355-360: I like the new results on the parameters for selection on UCNEs. I think this is a nice contribution to the paper. In addition to the number of sites under selection, might it also be the case that another unusual thing about UCNEs is that they are under selection in many species? This is reflected in the reduced divergence across species. Maybe consider adding a sentence or two about this?

We have added a brief mention of this point to the discussion about UCNEs.

Lines 370: When analyzing the Kim et al. DFE, the authors write, “For example, the best-fitting such model in a representative recent study by Kim et al. [8], based on a fairly large sample size (432 Europeans from the 1000 Genomes Project), implied a mean selection coefficient against amino-acid replacements of $s = 0.007$, corresponding to only $s_{het} = 1/2s = 0.0035$, since this model assumed additivity.” I think there is a factor of 2 error here. The Kim et al. DFE, as well as FitDadi and the previous PRF models (Williamson et al 2004 & Williamson et al. 2005; Boyko et al. 2008) parameterize the fitnesses as 1, $1+s$, and $1+2s$. Thus, the value of s that is inferred from these SFS-based methods corresponds to the fitness effect in the heterozygote, not the homozygote. Thus, the authors seem to be incorrectly dividing the scale parameter of the gamma DFE from Kim et al. by 2 in their analysis. This would result in them simulating less selection than inferred by Kim et al., perhaps accounting for some of the discrepancy in the models. The authors should investigate this more and adjust the conclusions about the differences in DFEs appropriately.

We carefully reviewed the work by Kim et al. and our own scripts and have determined that the reviewer is correct—we had made a factor of 2 error in our treatment of this issue. However, we had also been erroneously assuming a dominance coefficient of 1 rather than 0.5 in our simulations, which largely compensated for our other error. In any case, we have redone the simulations using the correct values and updated Table S2 and Figure S9 accordingly. Our results have not qualitatively changed, so we left the discussion largely the same, but we did correct the selection coefficient cited there and soften the language somewhat (“suggests that the SFS-based methods have under-estimated the weight of the tail” rather than “strongly suggests that the SFS-based methods have systematically under-estimated the weight of the tail”).

Table S2: Examination of the parameters here support my point above—the scale parameter for the gamma DFE in terms of the heterozygous fitness effect should be 0.03314752. Also, I believe there is a typo here for the shape parameter. It should be 0.199 and not 0.1930 as reported in the table.

The reviewer is correct about these points also. We have updated the Table accordingly and made the corresponding changes to Figure S9.